# RIG-I like receptor sensing of host RNAs facilitates the cell-intrinsic immune response to KSHV infection

Yang Zhao[1], Xiang Ye[1], William Dunker [1], Yu Song[1,2] & John Karijolich [1,3,4]

The RIG-I like receptors (RLRs) RIG-I and MDA5 are cytosolic RNA helicases best characterized as restriction factors for RNA viruses. However, evidence suggests RLRs participate in innate immune recognition of other pathogens, including DNA viruses. Kaposi's sarcoma-associated herpesvirus (KSHV) is a human gammaherpesvirus and the etiological agent of Kaposi's sarcoma and primary effusion lymphoma (PEL). Here, we demonstrate that RLRs restrict KSHV lytic reactivation and we demonstrate that restriction is facilitated by the recognition of host-derived RNAs. Misprocessed noncoding RNAs represent an abundant class of RIG-I substrates, and biochemical characterizations reveal that an infection-dependent reduction in the cellular triphosphatase DUSP11 results in an accumulation of select triphosphorylated noncoding RNAs, enabling their recognition by RIG-I. These findings reveal an intricate relationship between RNA processing and innate immunity, and demonstrate that an antiviral innate immune response can be elicited by the sensing of misprocessed cellular RNAs.

[1] Department of Pathology, Microbiology, and Immunology, Vanderbilt University School of Medicine, Nashville, TN 37232-2363, USA. [2] College of Pharmacy, Xinxiang Medical University, Xinxiang, Henan Province 453000, China. [3] Vanderbilt-Ingram Cancer Center, Nashville, TN 37232-2363, USA. [4] Vanderbilt Institute for Infection, Immunology and Inflammation, Nashville, TN 37232-2363, USA. Correspondence and requests for materials should be addressed to J.K. (email: john.karijolich@vanderbilt.edu)

The RIG-I-like receptor (RLR) family of PRRs is a group of cytosolic RNA helicases capable of discriminating self vs. nonself RNA. To date, three RLR members have been identified, retinoic acid-inducible gene-I (RIG-I; DDX58), melanoma-differentiation-associated gene 5 (MDA5), and laboratory of genetics and physiology 2 (LGP2)[1,2]. The RLRs share similar domain structures, including a central DExD/H-box helicase core, and a C-terminal domain that contributes to ligand discrimination. Additionally, within the N-terminus of RIG-I and MDA5 are two tandem caspase activation and recruitment domains (CARDs) that facilitate interactions with downstream adapter proteins. Upon ligand recognition, RIG-I and MDA5 undergo a series of conformational changes as well as post-translational modifications, which enables the oligomerization of CARDs[3,4]. RLR oligmerization facilitates their interactions with their common adapter mitochondrial antiviral-signaling protein (MAVS), leading to activation of the transcription factors interferon regulatory factor 3 (IRF3), IRF7, and an interferon gene-expression response[4].

MDA5 and RIG-I recognize distinct chemical and structural features of RNAs. The optimal substrate of MDA5 is long double stranded RNA (dsRNA) without unpaired bulged nucleotides[5]. In contrast, RIG-I senses short 5′ tri- and diphosphorylated dsRNA in which the terminal 5′- and 3′- nucleotides are base-paired[5–9]. Importantly, these features are generally absent from the host transcriptome. For instance, post-transcriptional adenosine-to-inosine editing in long-dsRNA structures disrupts long RNA secondary structure helices, thereby preventing immune activation[10–14]. Additionally, although RNAs are synthesized using triphosphates, the 5′ nucleotides are capped, or processed to remove the gamma- and beta-phosphate moieties, generating 5′-monophosphorylated RNAs[15].

As RNA sensors, restriction of RNA viruses, elicited by either viral genome or replicative intermediates, has been most extensively characterized[8,16–24]. Interestingly, several DNA viruses have also been reported to activate the RLR pathway. For example, Herpes simplex virus 1 (HSV-1) is recognized by both RIG-I and MDA5[25,26]. Interestingly, in HSV-1 infected HEK293T cells, host-derived 5S ribosomal RNA (rRNA) pseudogene RNA was defined as a prominent RIG-I ligand[27]. During Epstein–Barr virus (EBV) infection, EBV-encoded small RNAs are recognized by RIG-I[28–30]. Furthermore, poly dA:dT DNA sequences within some DNA viral genomes recruit cytoplasmic RNA polymerase (RNAP) III to facilitate transcription of short triphosphorylated noncoding RNAs which are recognized by RIG-I[31,32].

Kaposi's sarcoma-associated herpesvirus (KSHV) is an oncogenic gammaherpesvirus and an AIDS-associated pathogen. KSHV is the etiological agent of KS and the B cell lymphoma, primary effusion lymphoma (PEL)[33,34]. RIG-I and MAVS were previously reported to restrict KSHV lytic reactivation, and the virus deploys multiple viral proteins to disrupt RIG-I activity during lytic infection[35–37]. For example, KSHV de novo infection deposits teguments proteins into the cytoplasm, of which ORF75 has been show to promote the deamidation of RIG-I partially disrupting its RNA-binding potential[37]. However, the role of MDA5 in KSHV lytic reactivation is not known. In addition, the in vivo substrates that elicit RLR activation have not been identified.

Here, we define the relevance of RIG-I and MDA5 in KSHV lytic reactivation. We demonstrate that both RLRs are capable of restricting KSHV lytic reactivation in an epithelial-cell line as well as in patient-derived PEL cells. Strikingly, MDA5 is a more robust restriction factor than RIG-I, as depletion of MDA5 promotes a more significant increase in viral gene expression and production of infectious virions than RIG-I depletion. Additionally, using in vivo formaldehyde crosslinking RNA immunoprecipitation

(IP) sequencing (fRIP-seq) we identified the in vivo substrates of RIG-I and MDA5 during lytic reactivation in PEL cells. Remarkably, only host-derived RNAs were significantly enriched by RIG-I and MDA5 fRIP-seq, indicating that the cell-intrinsic immune response against KSHV is initiated by the sensing of host RNAs. We further identified a defect in cellular noncoding RNA 5′-end processing that enables the accumulation of 5′-triphosphorylated (5′-ppp) noncoding RNAs resulting in their recognition by RIG-I. This study defines the in vivo substrates of RIG-I and MDA5 in an oncogenic DNA virus infection, and unexpectedly, demonstrates that recognition of misprocessed host noncoding RNAs activates the cell-intrinsic immune response. Thus, there is an intricate relationship between cellular RNA processing and cell-intrinsic immunity that the host is capable of leveraging during infection.

## Results

**Depletion of RLRs and MAVs enhances KSHV lytic reactivation.** RIG-I and MDA5 initiate an interferon gene expression response following the detection of specific dsRNA (Fig. 1a). Although RIG-I and MAVS have been demonstrated to participate in the restriction of KSHV lytic reactivation, a role for MDA5 has not been addressed[35,36]. We hypothesized that both RIG-I and MDA5 restrict KSHV lytic reactivation and do so by recognizing distinct RNAs. To test our hypothesis, we first determined the contribution of each RLR sensor by depleting RIG-I, MDA5, or their adapter MAVS, individually in the KSHV-positive cell line iSLK.219. iSLK.219 cells contain a latent version of the KSHV genome expressing a constitutive green fluorescent protein (GFP) marker, and a doxycycline (Dox)-inducible version of the major viral lytic transactivator protein, RTA, to enable entry into the lytic cycle. The viral genome also contains a red fluorescent protein (RFP) marker driven by a lytic cycle-specific promoter, which can be used to monitor lytic reactivated cells. siRNA knockdown of both MDA5 and MAVS resulted in a striking increase in RFP positive cells 48 h postreactivation (Fig. 1b, c). In contrast, RIG-I depletion resulted in a modest increase in RFP positive cells over the control siRNA. To quantify lytic reactivation, we measured viral gene expression in latent and lytic iSLK.219 cells by reverse transcription quantitative polymerase chain reaction (RT-qPCR). While knockdown of RIG-I, MDA5, or MAVS did not promote spontaneous reactivation (Supplementary Fig. 1a), upon lytic reactivation, expression of the viral lytic genes ORF57, PAN, ORF52, ORF50 (RTA), ORF59, and bZIP were significantly increased relative to control siRNA treated cells (Fig. 1d, Supplementary Fig. 1b). Furthermore, western blot analyses demonstrated that viral ORF50 (RTA), ORF59, and bZIP expression was increased (Fig. 1e). We also determined the contribution of each RLR sensor and MAVS to KSHV de novo infection of iSLK cells. In contrast to what we observed during lytic reactivation, individual depletion of RIG-I, MDA5, or MAVS, only slightly increased KSHV de novo infection (Supplementary Fig. 1c–e).

Infectious viral progeny is only produced during the lytic cycle, thus we determined that effect of depleting RIG-I, MDA5, and MAVS on virion production using a supernatant transfer assay. iSLK.219 cells were treated with individual siRNAs and reactivated for 72 h, whereupon supernatants were collected and used to infect HEK293T recipient cells. Consistent with the observed increase in viral gene expression, depletion of both RLRs and MAVS resulted in a significant increase in the number of latent GFP-positive HEK293T cells (Fig. 1f, g). We quantified latent infection by measuring the expression of the viral latent gene latency associated nuclear antigen (LANA). MDA5 and MAVS depletion resulted in a greater than 20-fold

increase in LANA expression, whereas RIG-I resulted in a twofold increase (Fig. 1h). Together these results demonstrate that RIG-I and MDA5 restrict lytic reactivation and virion production and suggest that MDA5 is a more potent restriction factor.

Activation of RIG-I and MDA5 lead of an interferon gene expression response, thus we monitored IRF3 activation by subjecting latent and lytic iSLK.219 cell lysates to immunoblotting for phosphorylated IRF3. We observed IRF3 phosphorylation in siRNA control-treated cells, however, there was a complete loss of

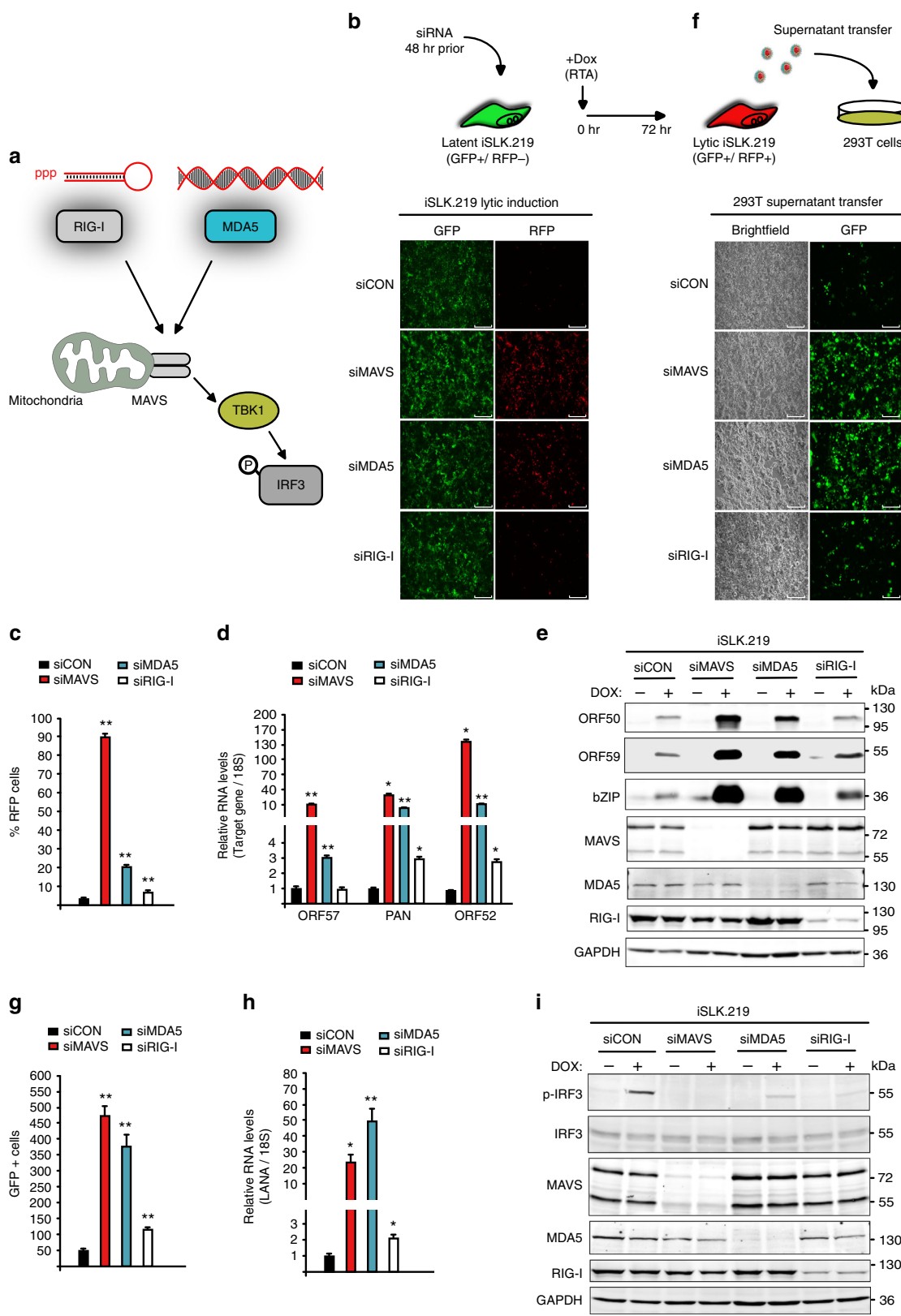

IRF3 phosphorylation in siRNA MAVS (Fig. 1i). Consistent with both MDA5 and RIG-I contributing to IRF3 phosphorylation, phosphorylated-IRF3 was reduced in both si-MDA5 and si-RIG-I treated cells.

**Overexpression of RLRs restricts KSHV lytic reactivation**. We next sought to determine whether expression of RIG-I or MDA5 would restrict KSHV reactivation. To test this, we transduced iSLK.219 cells with lentivirus harboring Dox-inducible FLAG-tagged RIG-I (F-RIG-I), MDA5 (F-MDA5), or a control empty cassette (Con), thus the addition of dox to the cell culture media reactivates KSHV as well as induces expression of F-RIG-I and F-MDA5. Reactivation of iSLK.219 cells transduced with F-RIG-I and F-MDA5 resulted in fewer RFP positive cells compared to cells transduced with the empty cassette (Fig. 2a). Quantification of viral reactivation by RT-qPCR demonstrated that the expression of viral lytic genes ORF57, PAN, vIL6, and ORF52 were all significantly reduced in F-RIG-I and F-MDA5 cells relative to the control (Fig. 2b). In addition, western blot analyses demonstrated reduced expression of ORF50, ORF59, and bZIP in F-RIG-I and F-MDA5 expressing cells compared to the control (Fig. 2c). Expression of MDA5 reduced both RNA and protein levels to a greater extent than RIG-I, supporting our knockdown data that MDA5 is a more potent KSHV restriction factor than RIG-I.

Next, using the established supernatant transfer assay, we monitored the production of infectious virions in the media of F-RIG-I, F-MDA5, and control cells 72 h postreactivation. There was a striking reduction in the number of GFP-positive cells when media from F-RIG-I and F-MDA5 was used to infect target HEK293T cells relative to media from control cells (Fig. 2d, e). Furthermore, quantification of LANA expression by RT-qPCR demonstrated an >80% reduction in LANA expression in cells infected with media from RLR-receptor overexpressing cells (Fig. 2f). Collectively, these results conclude that both RLRs restrict KSHV lytic reactivation in iSLK.219 cells and indicate MDA5 is a more robust restriction factor.

**RLRs and MAVS restrict KSHV lytic reactivation in PEL**. KSHV is the etiological agent of PEL, an aggressive HIV-associated non-Hodgkin's B cell lymphoma. We investigated the impact of RIG-I, MDA5, and MAVS on KSHV lytic reactivation in BC-3 cells, a patient-derived PEL cell line. We transfected BC-3 cells with shRNA expression constructs targeting RIG-I, MDA5, MAVS, or a control scramble sequence, and quantified their expression by RT-qPCR. All gene-targeting shRNAs significantly reduced their target gene expression by ~50% (Fig. 3a). To quantify the effect of their depletion on lytic reactivation, we reactivated BC-3 cells using sodium butyrate (NaB) and tetradecanoyl phorbol acetate (TPA), and measured expression of ORF50, ORF52, and PAN by RT-qPCR. Similar to our observed results in iSLK.219 cells, depletion of RIG-I, MDA5, and MAVS resulted in a significant increase in viral gene expression (Fig. 3b).

We reasoned that expression of RIG-I and MDA5 should restrict KSHV reactivation in PEL cells, and thus we established Dox-inducible F-RIG-I, F-MDA5, and control BC-3 cells. Reactivation of these cells with NaB, TPA, and Dox resulted the expression of Flag-tagged RIG-I and MDA5 (Fig. 3c). PAN RNA fluorescence in situ hybridization and flow cytometry (FISH-FLOW) determined that approximately 40% of the cells initiated lytic gene expression (Supplementary Fig. 2a). We next examined the levels of the viral proteins ORF50, ORF59, vIRF1, and bZIP by immunoblot analyses. All four viral proteins were expressed at reduced levels in cells expressing F-RIG-I and F-MDA5 relative to cells transduced with a control vector (Fig. 3c). Quantification of viral gene expression by RT-qPCR demonstrated a significant reduction in the expression of ORF57, ORF52, PAN, and vIL6 (Fig. 3d). Moreover, and consistent to what was observed in iSLK.219 cells, MDA5 displayed greater antiviral activity than RIG-I. This is likely a result of KSHV-mediated inhibition of RIG-I activity via deamidation of the RIG-I RNA-binding pocket, which reduces its affinity for RNA (Supplementary Fig. 2b–d). Finally, consistent with RIG-I and MDA5 restricting KSHV reactivation via the interferon pathway, the levels of phosphorylated IRF3 were increased in F-RIG-I and F-MDA5 expressing cells relative to the control (Supplementary Fig. 2e, f). Collectively, these results establish that RIG-I, MDA5, and MAVS participate in host defense against KSHV in PEL.

**RLRs bind host RNAs during lytic reactivation in PEL**. RIG-I and MDA5 restrict KSHV lytic reactivation in PEL cells and thus we reasoned that during KSHV lytic reactivation dsRNAs must be present. To test whether immunostimulatory dsRNAs are present in lytic BC-3 cells, we isolated total RNA and subjected it to IP using the dsRNA-specific antibody J2 or control IgG. Immunoprecipitated RNAs were then transfected into a human HCT116 colorectal carcinoma cell line containing a stably integrated interferon stimulated gene 54 (ISG54)-inducible luciferase reporter, and luciferase levels were determined 24 h post-transfection (Fig. 4a). J2-enriched RNAs from lytic BC-3 cells resulted in a significant increase in luciferase levels compared to RNAs isolated from latent BC-3 cells. These results are consistent with the presence of immunostimulatory dsRNA in lytic BC-3 cells.

Given that both RIG-I and MDA5 restrict KSHV reactivation in PEL, BC-3 cells provide an ideal system to define and compare the in vivo ligands of both RLRs in a physiologically relevant cell type. To identify the in vivo RNAs that are recognized by RIG-I and MDA5 and prevent postlysis RNA–protein interactions, lytic F-RIG-I, F-MDA5, and control BC-3 cells were formaldehyde cross-linked. Cell lysates were then subjected to a stringent purification protocol including washes in 1 M urea, prior to cross-link reversal and isolation of bound RNAs, and RNA sequencing (fRIP-seq). Remarkably, despite observing robust viral gene expression in the input sequencing libraries, our bioinformatics analyses revealed only the presence of host-derived RNAs among

**Fig. 1** Knockdown of RLRs and MAVS enhances KSHV lytic reactivation in iSLK.219 cells. **a** Schematic of RLR-MAVS signaling pathway. **b** iSLK.219 cells were transfected with indicated siRNAs for 48 h and then treated with Dox for 72 h. GFP and RFP were imaged 48 h post-Dox treatment. Bar indicates 750 μm. **c** RFP positive cells were quantified by flow cytometry 48 h post-Dox treatment. **d** RNA extracted from iSLK.219 24 h post-reactivation and expression of the indicated genes was quantified by RT-qPCR. **e** Western blot analysis of cell lysate from latent and 48 h post-Dox treatment iSLK.219 cells described in (**b**). **f** HEK293T cells were infected with supernatants of reactivated iSLK.219. GFP images were captured 48 h postinfection. Bar indicates 300 μm. **g** Quantification of GFP positive cells in (**f**). **h** RNA was extracted from HEK293T cells in (**f**) and KSHV LANA gene expression was monitored by RT-qPCR. **i** Western blot analysis of cell lysates from latent and 24 h post-Dox treatment iSLK.219 cells in (**b**). Error bars in all panels represent mean ± SD from three independent experiments. p Values were determined by the Student's t test, $^*p < 0.05$, $^{**}p < 0.01$

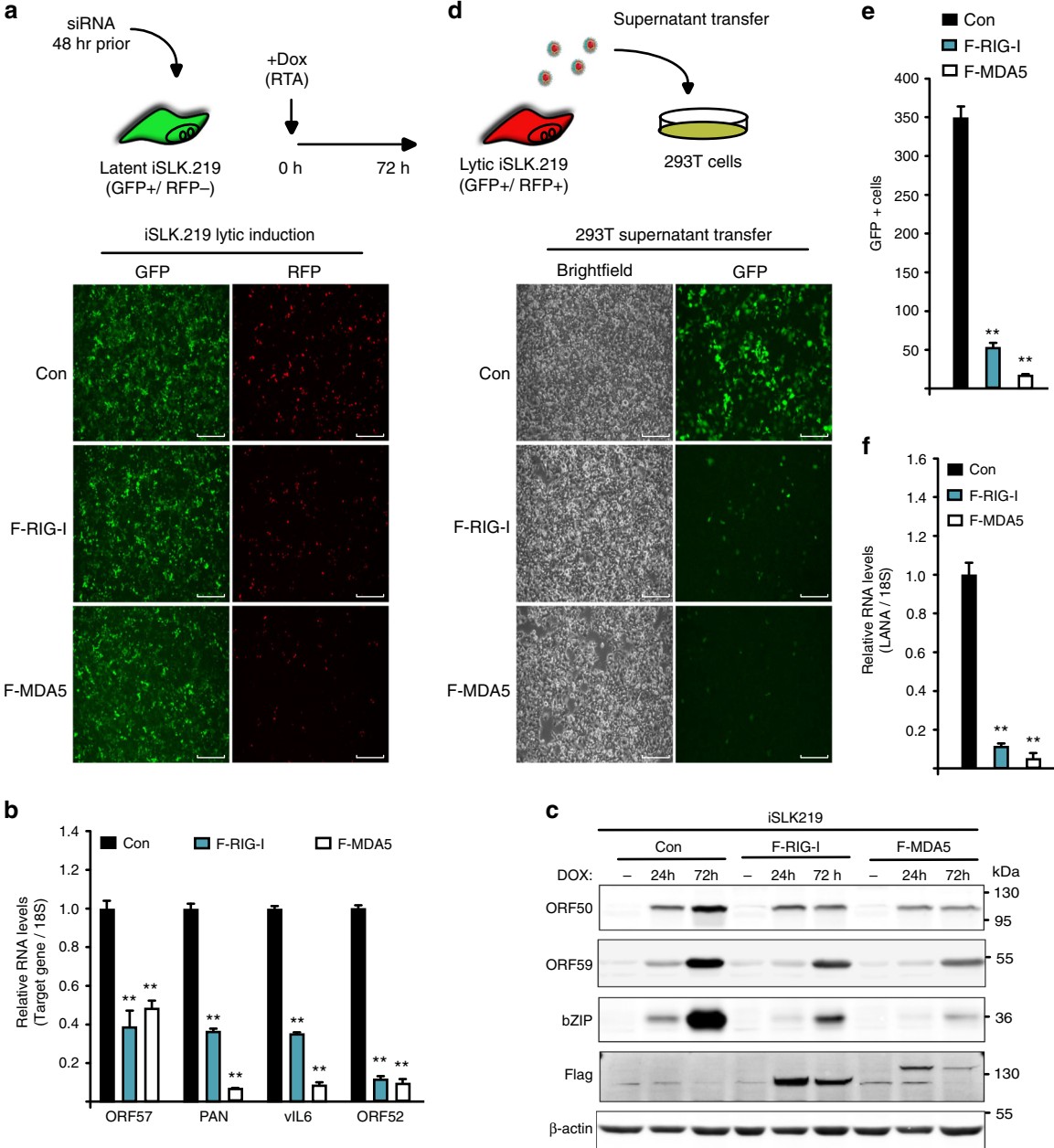

**Fig. 2** Overexpression of MDA5 or RIG-I restricts KSHV lytic reactivation in iSLK.219 cells. **a** iSLK.219 F-RIG-I, F-MDA5, and Control (Con) cells were reactivated for 72 h at which time GFP and RFP images were captured. Bar indicates 750 μm. **b** RNA was extracted from cells in (**a**) and expression of the indicated viral genes was quantified by RT-qPCR. **c** Western blot analysis of cell lysates prepared from cells described in (**a**). **d** HEK293T cells were infected with supernatants from cells described in (**a**). GFP images were taken 48 h postinfection. Bar indicates 300 μm. **e** Quantification of GFP-positive cells in (**d**). **f** RNA was extracted from HEK293T cells in (**d**). Expression of KSHV LANA was quantified by RT-qPCR. Error bars in all panels represent mean ± SD from three independent experiments. *p* Values were determined by the Student's *t* test, *$p < 0.05$, **$p < 0.01$

the significantly enriched F-RIG-I and F-MDA5 RNAs (Fig. 4b, Supplementary Fig. 4a, b). We identified 1324 and 650 bound RNAs for MDA5 and RIG-I, respectively, and the RNAs belong to both coding and noncoding biotype annotations (Fig. 4b).

The substrate specificity of RIG-I and MDA5 is best characterized in vitro and it has been determined that they recognize distinct chemical and structural features of dsRNA. We thus determined the similarity amongst F-RIG-I and F-MDA5 bound RNAs by principal component analysis (PCA). Indeed, consistent with expectations derived from in vitro binding experiments, PCA demonstrated F-RIG-I and F-MDA5 enriched

distinct RNAs from in vivo (Fig. 4c). Furthermore, consistent with PCA analysis, gene ontology analysis of the enriched RNAs demonstrated differential enrichment of ontologies (Fig. 4d, e).

We verified select F-RIG-I and F-MDA5 enriched RNAs by fRIP RT-qPCR. For instance, the NOP14 locus is significantly enriched in the F-MDA5 fRIP-seq data (Fig. 4f). We performed fRIP RT-qPCR and as expected NOP14 RNA was selectively immunoprecipitated by MDA5 (Fig. 4g). In addition, we verified GINS1 RNA is preferentially enriched by MDA5 (Fig. 4g). Among the most highly enriched RIG-I bound RNAs were the cellular vault RNAs (vtRNAs). There are four vtRNA genes

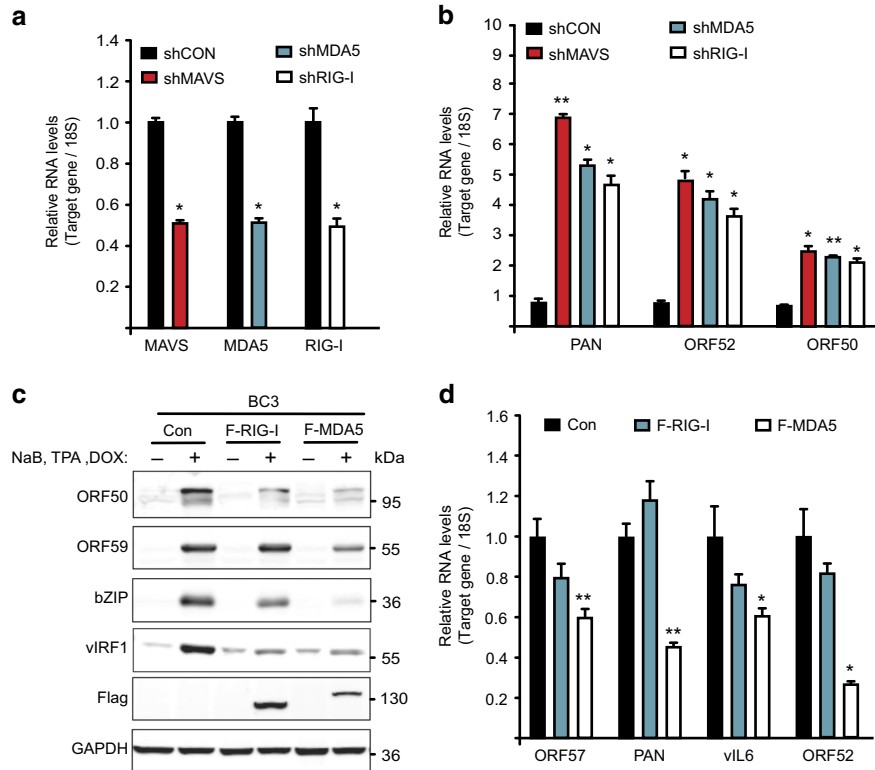

**Fig. 3** RLRs-MAVS signaling pathway restricts KSHV lytic reactivation in BC-3 cells. **a** RNA was extracted from BC-3 cells expressing the indicated shRNAs and the target mRNA level was quantified by RT-qPCR. **b** Expression of KSHV viral genes in cells described in (**a**) was quantified by RT- qPCR. **c** BC-3 F-RIG-I, F-MDA5, and Control (Con) cells were reactivated for 48 h. Cell lysates from latent and lytic cells were analyzed by immunoblotting for the indicated proteins. **d** RNA was extracted from BC-3 cells described in (**c**) and expression of the indicated viral genes was quantified by RT-qPCR. Error bars in all panels represent mean ± SD from three independent experiments. p-values were determined by the Student's *t* test, $^*p < 0.05$, $^{**}p < 0.01$

present in the human genome and vtRNAs 1–1, 1–2, and 1–3 were significantly enriched in the F-RIG-I fRIP-seq data (Fig. 4h)[38]. To verify that vtRNAs are indeed RIG-I substrates we performed fRIP RT-qPCR. Consistent with the fRIP-seq data, vtRNAs were selectively enriched in RIG-I eluates compared to eluates from MDA5 and control cell IPs (Fig. 4i).

**vtRNAs are RIG-I enriched immunostimulatory RNAs.** As the chemical specificity of RIG-I is more well biochemically defined we focused our subsequent analyses on defining the basis by which endogenous RNAs are rendered RIG-I substrates. The optimal RIG-I substrate is a short 5′ tri- and diphosphorylated dsRNA in which the terminal 5′- and 3′-nucleotides are base-paired. With the exception of a terminal U tail, RNA secondary structure predictions suggest vtRNAs 1–1, 1–2, and 1–3 adopt a similar conformation (Fig. 5a). RT-qPCR quantification of vtRNA expression during BC-3 lytic reactivation demonstrated vtRNAs 1–1 and 1–3 increased ~twofold, whereas vtRNA 1–2 slightly decreased (Supplementary Fig. 4a, b, c). Considering vtRNAs are enriched by RIG-I during lytic reactivation, we next tested whether transfection of vtRNAs is sufficient of elicit an interferon response, and whether the response is dependent on a 5′-triphosphate moiety. We in vitro transcribed vtRNAs 1–1, 1–2, and 1–3 and subjected a fraction of it to CIP treatment to remove the 5′-triphosphates before transfecting them into the HCT116 ISG54-luciferase reporter cell line (Fig. 5b, c, Supplementary Fig. 5). vtRNAs robustly stimulated the ISG54-luciferase reporter only when possessing a 5′-triphosphate (CIP-), consistent with them being recognized by RIG-I (Fig. 5c).

**DUSP11 reduction facilitates accumulation of 5′-ppp-vtRNA.** In unstressed cells vtRNAs are not immunostimulatory as their 5′-triphosphate moieties are removed by the cellular triphosphatase dual specificity phosphatase 11 (DUSP11)[39,40]. We hypothesized the immunostimulatory nature of vtRNA during lytic reactivation was because of a reduction in DUSP11 enzymatic activity or expression, resulting in an accumulation of 5′-ppp-RNAs. To test this hypothesis, we first determined whether DUSP11 expression was affected by KSHV lytic reactivation. Indeed, RT-qPCR and western blot analyses demonstrated that both the mRNA and protein level of DUSP11 were reduced by ~80% in lytic BC-3 cells relative to latency (Fig. 5d, e, Supplementary Fig. 6). This effect is likely mediated at the level of transcription, as we observed a reduction of RNAP II at the DUSP11 locus by chromatin IP qPCR (Fig. 5f).

Next, we tested whether vtRNAs are triphosphorylated during lytic reactivation. To test this, we developed a splint-ligation assay that leverages the ability of T4 DNA ligase to use a monophosphorylated RNA in a ligation reaction, while discriminating against a triphosphorylated substrate (Supplementary Fig. 7a, b). To quantify the amount of 5′-ppp-vtRNA we preformed splint-ligations on total RNA isolated from latent and 3-days postlytic reactivated BC-3 cells. To quantify the total amount of vtRNA in either condition, a fraction of the RNA was dephosphorylated and subsequently monophosphorylated with ATP prior to splint-ligation (Fig. 5g). Our splint-ligations assays demonstrated an ~50% reduction in splint-ligated products when using RNA from lytic BC-3 cells when compared to RNA from latent BC-3 cells, demonstrating an accumulation of 5′-ppp-vtRNAs upon lytic reactivation. Furthermore, we purified vtRNAs and

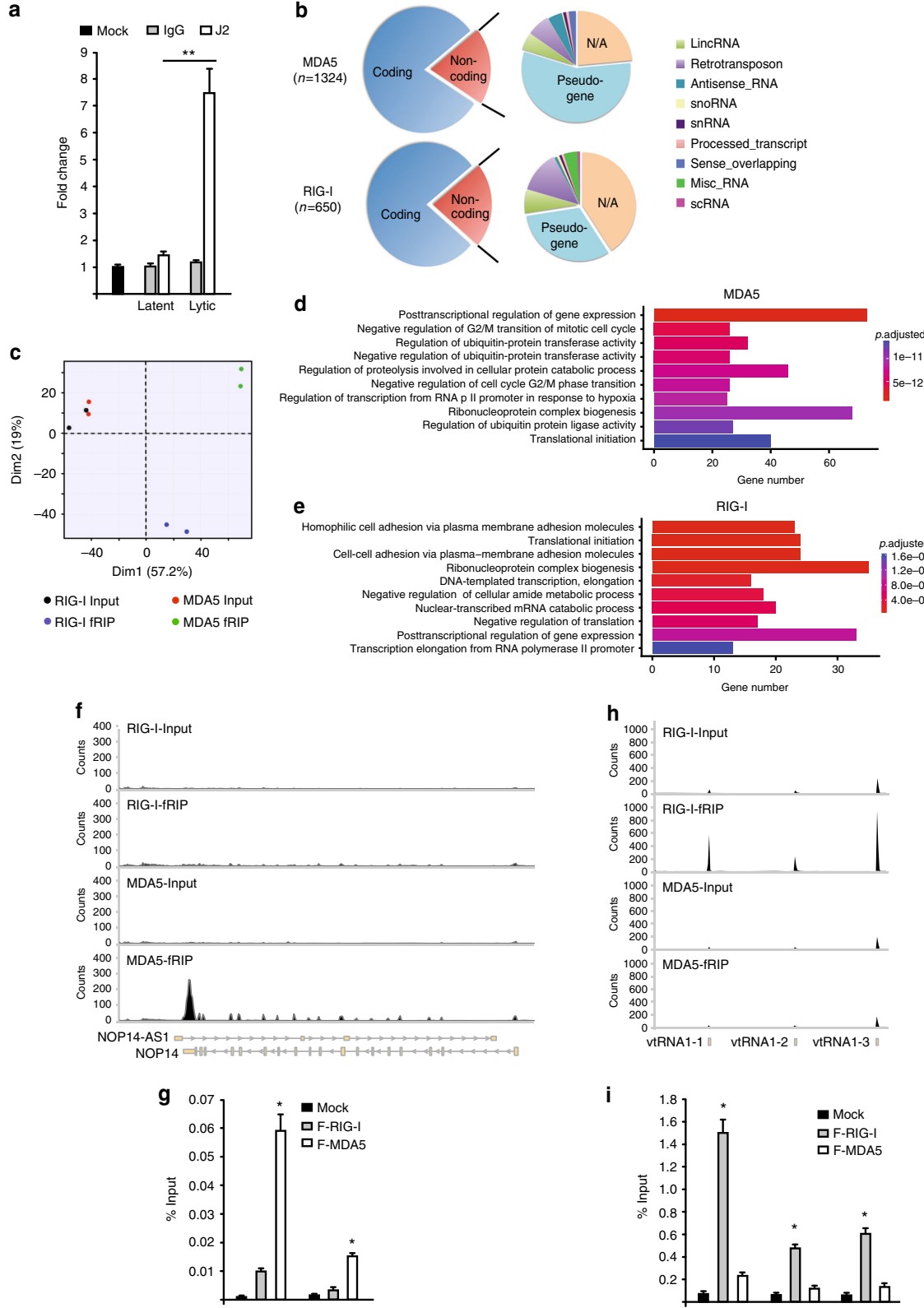

Fig. 4 RIG-I and MDA5 bind host RNAs during lytic reactivation in PEL. **a** HCT116 dual cells were transfected with RNA purified from latent and lytic BC-3 cells either by IgG or J2 antibody. Cells were harvested 24 h posttransfection and subjected to luciferase assay. Mock indicates no antibody. **b** Pie chart representation of gene biotypes identified by F-RIG-I and F-MDA5 fRIP-seq. **c** PCA analysis. **d**, **e** The gene ontology overrepresentation analysis of enriched RNAs from F-MDA5 (**d**) and F-RIG-I (**e**) fRIP-seq. **f** Distribution of MDA5 fRIP-seq reads on the NOP14 locus. **g** fRIP-qPCR analysis of NOP14 and GINS. **h** Distribution RIG-I fRIP-seq reads at the vtRNA loci. **i** fRIP-qPCR analysis of vtRNAs. Error bars in all panels represent mean ± SD from three independent experiments. $p$ Values were determined by the Student's $t$ test, $^*p < 0.05$, $^{**}p < 0.01$

U1 spliceosomal small nuclear RNA, a 2,2,7 trimethylguanosine capped RNA and thus not a DUSP11 substrate, from latent and lytic BC-3 cells using antisense oligonucleotide selection and transfected the purified RNA into HCT116 ISG54-luciferase reporter cells (Fig. 5h). While vtRNA purified from latent BC-3 cells did not induce luciferase expression, cells transfected with vtRNAs purified from lytic BC-3 cells had increased luciferase levels over control U1 snRNA transfected cells. Furthermore, CIP

treatment of vtRNA purified from lytic BC-3 resulted in a loss of immunostimulatory activity (Fig. 5h). Finally, we tested whether DUSP11 depletion, and the resultant increase in triphosphoryated RNAs, is sufficient to induce an interferon response outside the context of infection. Indeed, siRNA-mediated depletion of DUSP11 in HCT116 ISG54-luciferase cells induced luciferase 3-fold over a control siRNA (Supplementary Fig. 8a, b). Furthermore, transfection of 5′-ppp-RNA into DUSP11-

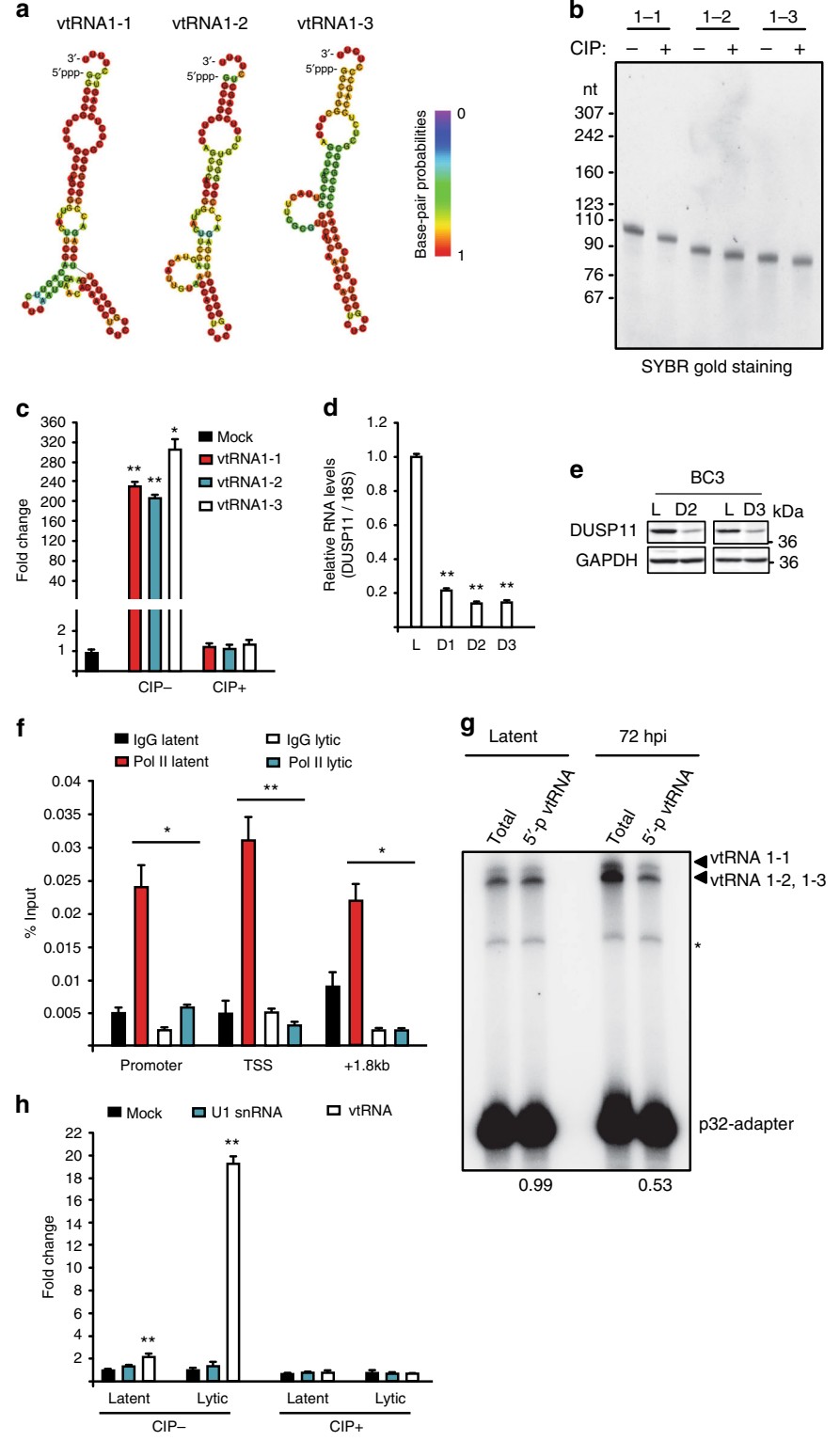

depleted HCT116 ISG54-luciferase cells resulted in a significant increase in luciferase levels over si-Control treated cells (Supplementary Fig. 8b).

**5′-ppp-vtRNAs block KSHV lytic reactivation.** We next tested whether transfection of 5′-ppp-vtRNAs can restrict KSHV lytic reactivation. In vitro transcribed 5′-triphosphate bearing and CIP-treated vtRNAs were transfected into iSLK.219 cells and lytic reactivation was induced 4 h later. RFP positive cells and viral gene expression was quantified 48 h after reactivation. In addition, we performed mock transfections, as well as transfected a well-defined commercially available RIG-I agonist as controls. While transfection of CIP-treated vtRNAs had minimal effect on viral gene expression and lytic reactivation, transfection of 5′-triphosphate bearing vtRNAs and RIG-I agonist significantly reduced both (Fig. 6a–c). Interestingly, transfection of 5′-triphosphate bearing vtRNAs and RIG-I agonist resulted in a minor increase in PAN, ORF52, and ORF57 expression in latent cells. This increase pales in comparison to lytic gene expression observed upon Dox-induced reactivation, and accordingly did not result in the presence of RFP-positive cells (Supplementary Fig. 9).

We quantified the expression of interferon inducible genes in vtRNA transfected iSLK.219 cells by RT-qPCR. The interferon inducible genes ISG15, IFI44, and IFIT2 were modestly induced in lytic mock- and CIP-treated vtRNA transfected cells. In contrast, 5′-triphosphate bearing vtRNA and RIG-I agonist significantly increased the expression of all measured interferon inducible genes (Fig. 6d). Collectively, these results demonstrate that 5′-triphosphate bearing vtRNAs are immunostimulatory and restrict KSHV lytic reactivation.

## Discussion

RLR-receptors elicit an interferon gene expression response upon recognition of specific nucleic acid features that are typically associated with RNA viral pathogens. While in vitro experiments have defined these signatures, few studies have defined RNAs that are sensed in vivo. Here, we have determined the contribution of the RLR-sensing pathway to KSHV lytic reactivation as well as defined the in vivo ligands of RIG-I and MDA5. Our data demonstrate that RIG-I and MDA5 restrict KSHV lytic reactivation in both iSLK.219 and patient-derived PEL BC-3 cells. Unexpectedly, RNAs bound to RIG-I and MDA5 during KSHV lytic reactivation are host-derived. Consistent with RIG-I and MDA5 sensing distinct features of RNAs in vitro, bioinformatics analyses demonstrate minimal overlap between RNAs recognized by RIG-I and MDA5 in vivo. Our work identifying triphosphorylated RNAs bound to RIG-I puts forth a model whereby RIG-I dependent sensing is triggered upon an infection-dependent reduction in the expression of the RNA triphosphatase DUSP11, resulting in an accumulation of triphosphorylated RNAs (Fig. 6e).

Both MDA5 and RIG-I contribute to the restriction of KSHV. Interestingly, despite MDA5 having a more pronounced antiviral effect, depletion of RIG-I or MDA5 results in a similar reduction in IRF3 phosphorylation. This finding suggests an unidentified mechanism of antiviral activity that MDA5 possesses that is independent of its canonical signaling pathway. Indeed, it is becoming increasingly clear that both the RNA and DNA sensing machinery exert antiviral activity outside their canonical pathways. For instance, RIG-I has been shown to directly interact with and activate components of the JAK/STAT pathway[41,42]; while the DNA sensor STING can exert antiviral activity toward RNA viruses via the inhibition of translation[43]. The elucidation of this additional pathway by which MDA5 can influence antiviral responses will likely yield new insights into host–pathogen interactions.

Our findings demonstrating that both RIG-I and MDA5 sense host-encoded RNAs during KSHV reactivation is remarkable, and indicates that antiviral immunity can be triggered through the sensing of host RNAs. This is consistent with recent data demonstrating RNase L cleavage products and exosomal 7SL RNA can serve as RIG-I ligands[44,45]. Furthermore, the most prominent RIG-I bound RNA in HSV-1 infected HEK293T cells is host-encoded 5S ribosomal RNA pseudogene 141 (RNA5SP141)[27]. However, RNA5SP141 is unlikely to be a universal activator of the RIG-I pathway following viral infection. In fact, RNA5SP141 is not expressed in PEL, or present in RIG-I or MDA5 fRIP-seq data from PEL cells. Interestingly, however, we do detect pseudogene transcripts derived from other RNAs, including spliceosomal U6 small nuclear RNA, an RNAP III transcribed RNA. Pseudogenes are known to participate in the posttranscriptional regulation of gene expression by functioning as competing endogenous RNAs[46], and it is interesting to speculate that pseudogenes have also been co-opted into condition-specific triggers of interferon responses.

RNA unshielding, loosely defined as an alternation in the stoichiometry of a triphosphorylated RNA and its given RNA-binding proteins, is implicated in the RIG-I sensing of exosomal 7SL RNA and RNA5SP141[27,45]. While the RNA-binding proteins of vtRNAs are not well-defined, in fact only ~5% of vtRNAs are present within the vault complex for which they are named, it is unlikely that RNA unshielding is the only contributor to RIG-I activation during KSHV reactivation. This is perhaps best exemplified by our finding that Y RNAs, which are components of Ro60 ribonucleoprotein particles, are recognized by RIG-I during reactivation, yet there is no decrease in the expression of Ro60 at the RNA or protein level. Furthermore, the levels of Y RNAs are relatively constant throughout reactivation. Thus, the

**Fig. 5** Accumulation of immunostimulatory 5′-ppp-vtRNAs during lytic reactivation. **a** Predicted secondary structure of vtRNAs generated by RNAfold. **b** SYBR-Gold staining of in vitro transcribed vtRNAs with or without CIP treatment. **c** HCT116 ISG54-luciferase reporter cells were transfected with 100 ng in vitro transcribed vtRNAs with or without CIP treatment. Cells were harvested 24 h posttransfection and subjected to luciferase assay. Mock indicated cells without RNA transfection and was set as 1. **d** BC-3 cells were reactivated for 3 days and expression of DUSP11 was quantified by RT-qPCR. L latency, D1–D3 lytic reactivation for 1 day to 3 days. The DUSP11 expression was normalized to the level of 18S rRNA and L was set as 1. **e** Cell lysates were prepared from BC-3 cells described in (**d**) and DUSP11 protein levels were monitored by Western blot. GAPDH was run as a loading control. **f** Latent and lytic BC-3 cells were subjected to RNAP II ChIP-qPCR analysis. Signals were normalized to input. **g** Total RNA, extracted from latent or 72 h postreactivation BC-3 cells, was subjected to splint-ligation to quantify 5′-monophosphorylated vtRNAs. * denotes a product of adapter-adapter ligation (see Supplementary Fig. 7). **h** HCT116 ISG54-luciferase reporter cells were transfected with vtRNA or U1 RNA isolated by antisense oligonucleotide affinity selection from either latent or lytic BC-3 cells. Cells were harvested 12 h posttransfection and subjected to luciferase assay. Mock indicated cells without RNA transfection and was set as 1. Error bars in all panels represent mean ± SD from three independent experiments. *p* Values were determined by the Student's *t* test, *p < 0.05, **p < 0.01

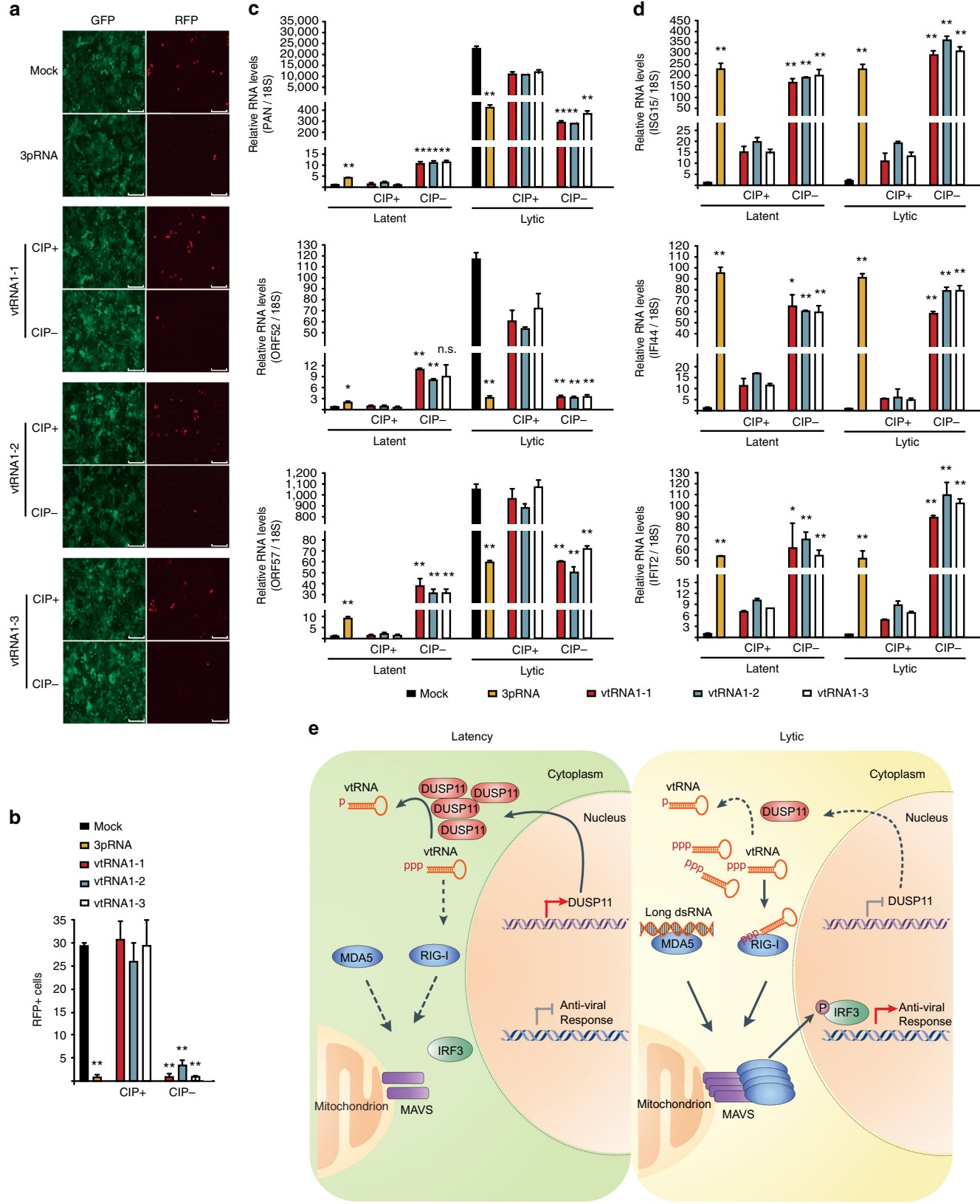

recognition of RNA by RIG-I in the context of KSHV reactivation is likely the result of multiple mechanisms, and dependent on the precise molecular steps involved in the biogenesis of a given RNA, with a deficiency in 5′-end processing one potential mechanism.

RNA processing is an essential step in gene expression and here we demonstrate a failure to properly process 5′-ends of vtRNAs contributes to their recognition by RIG-I. The extent to which defects in other RNA processing steps contribute to RIG-I

**Fig. 6** 5′-triphosphate containing vtRNAs block KSHV lytic reactivation. **a** iSLK.219 cells were mock transfected, or transfected with 100 ng in vitro transcribed vtRNAs with or without CIP treatment, or a RIG-I ligand RNA (3pRNA) and reactivated by adding Dox 4 h posttransfection. GFP and RFP images were captured 48 h postreactivation. Bar indicates 300 μm. **b** Quantification of RFP positive cells in (**a**). **c** Expression of the indicated viral genes was determined in latent and 48 h post-Dox treatment cells by RT-qPCR. **d** Expression of the indicated genes was quantified in latent and 24 h post-Dox treatment cells by RT-qPCR. **c**, **d** The gene expression was normalized to the level of 18S rRNA and Mock in latent cells was set as 1. Error bars in all panels represent mean ± SD from three independent experiments. p Values were determined by the Student's t test, $^*p < 0.05$, $^{**}p < 0.01$. **e** The model depicting how RLRs-MAVS pathway is activated and restricts KSHV lytic reactivation. In latency, DUSP11 removes 5′ triphosphates of vtRNAs, thus preventing their recognition by RIG-I. During lytic reactivation, 5′-end processing of vtRNAs is attenuated due a reduction in DUSP11 expression. 5′-ppp-vtRNAs and long dsRNAs are sensed by RIG-I and MDA5, respectively. RLRs elicit an antiviral gene expression program through MAVS and downstream phosphorylation of IRF3

activation is unclear, and identification of these molecular steps and defective processing intermediates may shed light into the molecular basis of disease, as well as represent a therapeutic target that can be leveraged. DUSP11 is an RNA triphosphatase that acts on RNAP III transcribed noncoding RNAs, thus it is likely that other DUSP11 substrate RNAs are also sensed[39]. In fact, Alu's, a family of noncoding retrotransposons transcribed by RNAP III that are DUSP11 substrates, are present in RIG-I fRIPs. This suggests that retrotransposon-derived RNAs are capable of being sensed by RLR-receptors and triggering an immune response. This would be consistent with reports demonstrating a role for these retrotransposon sequences in activation of immune responses in systemic lupus erythematosus, age related macular degeneration, and murine gammaherpesvirus 68 infection[47–50]. Given the role of DUSP11 in preventing the accumulation of endogenous RIG-I ligands it will be necessary to determine whether it is associated with autoimmune disease. In contrast, if molecules can be identified that reduce DUSP11 catalytic activity, promoting the activation of RIG-I, they may hold therapeutic promise for the treatment of some infectious diseases or even cancer.

RNAP III participates in cell-intrinsic innate defense to variety of pathogens, including DNA viruses[31,32]. The model for RNAP III triggered innate responses is that poly dA:dT DNA sequences within some DNA viral genomes are able to recruit cytoplasmic RNAP III, resulting in the transcription of short triphosphorylated noncoding RNAs which are then detected by RIG-I. We do not observe any significant enrichment of fRIP-seq reads that map to the KSHV genome and thus it is likely that RNAP III does not directly sense the KSHV genome. However, our identification of RNAP III transcribed host RNAs in RIG-I fRIP-seq data demonstrate RNAP III does participates in the innate immune response to KSHV by generating RNAs that can be sensed. Thus, RNAP III is a critical component of the cell-intrinsic immune response even in the absence of direct pathogen recognition.

Genome-wide association studies have revealed the association of single nucleotide polymorphisms (SNPs) in RIG-I and MDA5 with the risk of many autoimmune diseases, including systemic lupus erythematosus, Aicardi–Goutieres syndrome, and Singleton–Merten syndrome[51–53]. Some of these SNPs increase the sensitivity of the RLR-receptors, and thus they engage cellular RNAs leading to an interferon gene expression response. The in vivo ligands that are sensed and drive the interferon responses are just beginning to be identified[54]. Given all RLR-sensed ligands during KSHV infection are host-derived, these results should provide insight into potential substrates and mechanisms of RLR activation for several pathologies, not just KSHV infection.

## Methods

**Cells and viruses**. iSLK.219[55] (kindly provided by Dr. Britt Glaunsinger, University of California, Berkeley), iSLK.219 control, iSLK.219 FLAG-RIG-I, iSLK.219

FLAG-MDA5, and HEK293T (ATCC) were maintained in Dulbecco's modified Eagle medium (DMEM; Invitrogen) supplemented with 10% fetal bovine serum (FBS; Invitrogen). BC-3[56] (kindly provided by Dr. Britt Glaunsinger, University of California, Berkeley), BC-3 control, BC-3 FLAG-RIG-I, and BC-3 FLAG-MDA5 PEL cells were grown in RPMI 1640 medium (Invitrogen) supplemented with 10% FBS (Invitrogen) and 2 mM L-glutamine (Invitrogen). All cells were maintained with 100 U of penicillin/ml and 100 μg of streptomycin/ml (Invitrogen) at 37 °C under 5% $CO_2$. iSLK.219 and derivatives were reactivated with 1 μg/ml of doxycycline (Fisher Scientific), while BC-3 cells were reactivated using TPA (20 ng/ml; Sigma Aldrich) and NaB (0.1 mM; Sigma Aldrich).

**Cloning and lentivirus production**. RIG-I and MDA5 were PCR amplified from pEF-BOS-RIG-I and pEF-BOS-MDA5 and cloned into pDONOR221, and subsequently pLenti-CMVtight-FL-HA-DEST-Blast (Addgene) using Gateway cloning (Invitrogen). Lentivirus was prepared in HEK293T cells. Cells were transfected at 50–60% confluency with RIG-I, MDA5, or Empty destination vectors, psPAX2 (Addgene), and pMD2.G (Addgene) using polyjet (SignaGen). After 72 h post-transfection the supernatant was collected, adjusted to 8 μg/ml polybrene (Millipore), and target cells were spinfected at 1000 g for 1 h at room temperature. Cells were selected for 2 weeks in media containing 5 μg/ml blasticidin (Invivogen).

**Flow cytometry**. iSLK, iSLK.219, and its derivative cells were fixed with 2% paraformaldehyde and then analyzed on BD LSR Fortessa or Canto II instrument. Data were analyzed with FlowJo X software (TreeStar). The gating strategy is shown in Supplementary Fig. 11.

**Fluorescence in situ hybridization and flow cytometry**. FISH-FlOW of PAN RNA in BC-3 cells was done as previously described[57]. Briefly, latent or reactivated BC-3 and derivative cells were fixed in 4% (vol/vol) paraformaldehyde and permeablized with 1× PBS containing 0.2% (vol/vol) Tween-20. The permeabilized cells were then hybridized with Fluorescein-12-dUTP labeled PAN antisense oligos in HB 10% dx buffer (10% (wt/vol) dextran sulfate, 2× saline-sodium citrate (SSC), 10% (vol/vol) formamide, 1 mg/ml tRNA and 0.2 mg/ml BSA) at 37 °C overnight. After extensive washing with HBW buffer (2× SSC, 10% (vol/vol) formamide and 0.2 mg/ml RNase-free BSA), cells were analyzed on BD Canto II instrument. Data were analyzed with FlowJo X software (TreeStar). The gating strategy is shown in Supplementary Fig. 11.

**siRNA knockdowns**. iSLK.219 cells were transfected at 60–80% confluency with 40 nM siRNA (sequences in Supplementary Table 1) or MISSION siRNA Universal Negative Control #1 (Sigma) using Lipofectamine RNAiMax (Invitrogen). After 48 h posttransfection cells were reactivated as described above.

**shRNA knockdowns**. MISSION shRNA for MAVS (TRCN0000149206), RIG-I (TRCN0000152922), MDA5 (TRCN0000050852), and nontargeting shRNA (SHC016) were obtained from Sigma Aldrich. BC-3 cells were microporated with shRNA expression vectors using the Neon transfection system (Invitrogen) at 1400v, 10 ms pulse width, and 3 pulses. After 24 h postmicroporation cells were reactivated as described above.

**Supernatant transfer**. iSLK.219 and derivatives cells were reactivated with doxycycline for 72 h, after which the supernatant was collected, adjusted to 8 μg/ml polybrene, and HEK293T cells were spinfected at 1000 g for 1 h at room temperature. The infection media was replaced with fresh media and incubated for 72 h, followed by analysis.

**RT-qPCR**. Total RNA was isolated with TRIzol (Invitrogen) in accordance with the manufacturer's instructions. RNA was DNase I (NEB) treated at 37 °C for 20 min, and inactivated with EDTA at 70 °C for 10 min. cDNA was synthesized from DNase-treated RNA with random 9-mer (Integrated DNA Technologies) and M-

MLV RT (Promega). qPCR was performed using the PowerUp SYBR Green qPCR kit (Thermo Scientific) with appropriate primers (Supplementary Table 1).

**Prediction of RNA secondary structure**. Secondary structure analysis of vtRNAs was performed using RNAfold (http://rna.tbi.univie.ac.at/cgi-bin/RNAWebSuite/RNAfold.cgi) using default parameters.

**Antisense oligo affinity purification**. U1 snRNA and vtRNAs were isolated by antisense oligonucleotide affinity selection as previously described[58]. Briefly, RNA was isolated from ~200 million latent and lytic BC-3 cells. The RNA pellet was resuspended in 1 ml hybridization buffer (750 mM NaCl, 1% SDS, 50 mM Tris pH 7.0, 1 mM EDTA, 15% formamide and RNase inhibitor). Totally, 40 pmol of each 5′-TEG biotinylated (Supplementary Table 1) antisense oligonucleotide were added and the mixture was rotated end-over-end at 37 °C for 12 h. Totally, 100 μl of streptavidin-magnetic C1 beads were blocked with 100 μg/ml glycogen and 1 mg/ml BSA for 1 h at room temperature. Blocked C1 beads were added and the reaction was mixed for another 2 h at 37 °C. Complexes were captured by magnets (Invitrogen) and washed five times with wash buffer (2× SSC, 0.5% SDS, and RNase inhibitor). After the final wash, beads were resuspended in elution buffer (50 mM Tris pH 7.0, 100 mM NaCl, 1 mM EDTA). Beads in elution buffer were heated to 70 °C for 5 min before separating the elution buffer from the beads.

**Western blotting**. Whole cell lysates were prepared with lysis buffer (50 mM Tris [pH 7.6], 150 mM NaCl, 0.5% NP-40) and quantified by Bradford assay (BioRad). Equivalent amounts of each sample were resolved by sodium dodecyl sulfate polyacrylamide gel electrophoresis, electrotransferred to polyvinylidene difluoride membrane (Millipore), and blotted for the indicated proteins. Antibodies: GAPDH (Invitrogen, GA1R, #MA5-15738, diluted 1:5000), β-actin (Invitrogen, BA3R, #MA5-15739, 1:1000), vIRF1 (1:1000, kindly provided by Dr. Gary Hayward, John Hopkins University) ORF50 and ORF59 (1:10,000, kindly provided by Dr. Britt Glaunsinger, University of California, Berkeley), ORF57 (1:1000, kindly provided by Dr. Zhi-Ming Zheng, NCI), and bZIP (1:2000, kindly provided by Dr. Cyprian Rossetto, University of Nevada, Reno). IRF3 (Cell Signaling Technology, D83B9, #4302, 1:1000), Phospho-IRF3 S386 (Abcam, EPR2346, #ab76493, 1:1000), FLAG (Thermo Fisher Scientific, FG4R, MA1-91878, 1:1000), MAVS (Bethyl, #A300-782A, 1:1000), DDX58 (Abcam, EPR18629, #ab180675, 1:1000), MDA5 (Cell Signaling Technology, D74E4, #5321, 1:1000), DUSP11 (Proteintech, #10204-2-AP, 1:1000). Primary antibodies were followed by AlexaFluor 680-conjugated secondary antibodies (Life Technologies, goat anti-rabbit #A27042, goat anti-mouse #A28183, 1:10,000), and visualized by Odyssey CLx imaging system (LI-COR). The full size immunoblots scans are presented in Supplementary Fig. 10.

**In vitro transcription**. vtRNA in vitro transcription templates were generated by overlapping PCR of two DNA oligonucleotides (Supplementary Table 1). In vitro transcription was carried out at 37 °C overnight using the HiScribe T7 High Yield RNA Synthesis Kit (New England Biolabs), followed by DNase I digestion. In vitro transcription reactions were purified using Biospin 6 columns (BioRad) and fractionated on 7 M urea 8% polyacrylamide gels. Full-length vtRNAs were excised from the gel and RNAs were purified as previously described[58].

**Luciferase assays**. HCT116 Dual cells (Invivogen) were transfected with RNA using RNAiMAX (Invitrogen). Totally, 12–24 h posttransfection cell supernatants were collected and used to measure secreted Lucia luciferase activity using QUANTI-Luc (Invivogen) on a GLOMAX 20/20 Luminometer (Promega).

**Double stranded RNA IP**. dsRNA was immunoprecipitated using the dsRNA-specific J2 antibody. Briefly, 5 μg of J2 antibody or control IgG were added to total RNA and rotated overnight at 4 °C, whereupon 50 μl of prewashed protein G magnetic beads were added and rotated for an additional 2 h. Antibody-bead complexes were then isolated on a magnetic stand and washed three times with high-salt lysis buffer containing 500 mM NaCl. RNA was eluted by TRIzol extraction.

**Splint-ligation analysis**. For total vtRNA measurements, 50 μg of total RNA was treated with calf intestinal phosphatase (CIP) (NEB) for 1 h at 37 °C according to manufactures recommendation. RNA was extracted with phenol:chloroform:iso-amyl alcohol [25:24:1 (vol/vol)] followed by ethanol precipitation. A 5 μg of CIP-treated RNA was phosphorylated with ATP and T4 polynucleotide kinase prior to a biospin 6 (BioRad) clean up. Ligations were performed similar to as previously described[59]. For ligation, all reactions consist of 100fmol bridge oligonucleotide, 200fmol radiolabeled ligation oligonucleotide, 5 μg RNA, 8% PEG 8000, 1× T4 DNA ligase reaction buffer (NEB), and 10 units T4 DNA ligase (NEB). Before adding T4 DNA ligase, the reaction mixture was denatured at 95 °C for 1 min, cooled to 65 °C for 5 min, and 37 °C for 10 min, the ligase was added to the reaction mixture and incubated at 30 °C for 4 h. Reactions were terminated by heat inactivation at 75 °C for 10 min and subsequently separated using denaturing 8 M urea 10% polyacrylamide gels and imaged using a PhosphorImager.

**fRIP-seq**. fRIP-seq was performed as previously described with minor modifications[60]. Briefly, BC-3 cells were cross-linked with 1% formaldehyde for 10 min in PBS, and unreacted formaldehyde was neutralized with 0.3 M glycine for 5 min. Cells were washed 2× with PBS, then resuspended in RIPA buffer (50 mM Tris, pH 8.0, 1% IGEPAL CA 630, 0.5% sodium deoxycholate, 0.05% SDS, 1 mM EDTA, 150 mM NaCl, 1 mM DTT, and RNase and protease inhibitors) and kept on ice for 10 min. Soluble cell extracts were incubated with anti-FLAG M2 magnetic resin (Sigma) at 4 °C for 2 h. Resin was washed three times for 10 min and then two times for 5 min at room temperature in RIPA buffer containing 0.1% SDS, 1 M NaCl, and 1 M urea. Resin was eluted with 1×FLAG peptide in RIPA buffer for 45 min at 4 °C. Protein–RNA cross-links were reversed by adding 100 mM Tris, pH 8.0, 10 mM EDTA, 1% SDS, and 2 mM DTT to eluted samples and heating to 70 °C for 45 min. RNA was recovered by extraction with TRIzol and then again with phenol:chloroform:isoamyl alcohol [25:24:1 (vol/vol)] followed by ethanol precipitation. Paired-end RNA-sequencing libraries were prepared from the recovered RNA using the NEBNext Ultra II Directional RNA Library Prep Kit (NEB) according to the manufacture recommendations. Libraries were then subjected to paired-end sequencing on a HiSeq3000 with 150 cycles at the Vanderbilt Technologies for Advanced Genomics (VANTAGE).

**RNA-seq data analysis**. Raw read quality in fastq files were accessed by FastQC (www.bioinformatics. babraham.ac.uk/projects/fastqc). Reads were aligned to the human reference genome (gencode GRCh38.p10) and KSHV genome (GQ994935.1) using Spliced Transcripts Alignment to a Reference (STAR) software[61]. We estimated transcript and gene abundances, as well as depletion/enrichment significance using cufflinks and cuffdiff 2[62], and summarized to biotypes annotated in GENCODE database (gencode.v27s), remaining unannotated reads were further annotated using RepeatMasker annotation library hg38.fa.out.gz (http://www.repeatmasker.org). Statistical analyses and virtualization were performed using R packages: principle component analysis was performed using the PCA function in package FactoMineR[63] and presented with factoextra, genome coverages were presented with Gviz[64], the gene ontology overrepresentation analysis was performed with clusterprofiler on GO biological process[65].

**Reporting summary**. Further information on research design is available in the Nature Research Reporting Summary linked to this article.

## Data availability
Sequencing data from this study have been deposited in GEO under accession number GSE116650 [https://www.ncbi.nlm.nih.gov/geo/query/acc.cgi?acc = GSE116650]. A reporting summary for this article is available as a Supplementary file. All codes used in this study are available upon requested from the corresponding author.

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

## Acknowledgments

We would like to thank members of the Karijolich lab for insightful discussion. We thank Britt Glaunsinger (University of California, Berkeley,) Gary Hayward (John Hopkins University), Zhi-Ming Zheng (National Cancer Institute), and Cyprian Rossetto

(University of Nevada, Reno) for providing antibodies against KSHV proteins. This work was support by institutional startup funds (to J.K.), a grant from the Vanderbilt-Ingram Cancer Center Ambassadors program (to J.K.), American Cancer Society Institutional Research Grant (#IRG-58-009-58; to J.K.). J.K. is a Pew Biomedical Scholar.

## Author contributions

Conceived and designed the experiments: J.K. and Y.Z. Performed the experiments: Y.Z., W.D., X.Y., Y.S., and J.K. Analyzed the data: J.K., Y.Z., W.D., X.Y., and Y.S. Wrote the paper: J.K. and Y.Z.

## Additional information

**Competing interests:** The authors declare no competing interests.

