## [Peer Review File · Nature Communications]

Reviewers' comments:

Reviewer #1 (Remarks to the Author):

This submission makes three observations. The first is that RIG-I and MDA5 restrict Kaposi Sarcoma associated herpesvirus (KSHV) reactivation. The second observation is that vtRNAs are a new type of RIG-I substrate, which seems to support the notion that endogenous, rather than pathogen-derived molecules drive the RIG-I dependent interferon response to infection. The third observation is that vtRNAs are normally de-phosphorylated by DUSP11 and thus do not trigger RIG-I, but that this enzyme's expression is reduced in response to KSHV reactivation.

The first observation was previously published, and it is nice to see an independent confirmation of this result. The second observation is potentially transformative, but highly controversial. Most data attribute the activation of RIG-I directly to virus-derived products. This is certainly the case for any and all RNA virus infections. The notion that host RNAs trigger RIG-I in response to DNA virus infection is interesting, but in this submission originates from a failure to detect viral RNAs associated with RIG-I. The involvement of DUSP11 in the current process is not entirely novel, but really interesting. Unfortunately, the mechanistic studies are incomplete.

Finding RIG-I substrates is certainly an important goal for the field. Perhaps the study should be more focused on vtRNAs and DUSP11, for which the authors introduced novel biochemical assays. The connection of vtRNAs and DUSP11 with KSHV infection and potential physiological relevance is not as mature as it could be.

Specific points:

(1)

In addition to MDA5 and RIG-I, LGP2 also recognized foreign nucleic acids, but was not further assessed. Since Figure 1 has essentially been published before, it could be moved into supplemental data or enhanced with depletion of LGP2.

(2)

The experiments are based on forced expression of MDA5 and RIG-I. This is necessary to purify interacting RNAs, but it introduces a bias in ascertaining the biological relevance with regard to viral replication and reactivation. Overexpression is not very relevant physiologically, as RIG-I and MDA5 need to become activated before initiating interferon signaling. Overexpression may artificially drive oligomerization.

(3)

Does the forced overexpressing of RIG-I or MDA5 result in greater levels of IRF3 phosphorylation or are the limiting factors of the IFN response MAVS and IRF3 levels?

(4)

Figure 3: The WB analysis should include a late viral protein, which may show an even higher level of suppression by RIG-I or MDA5. In herpesviruses it is only after viral replication that all RNAs are transcribed including transcripts on opposite strands, which may give rise to double-stranded RNAs.

(5)

Figure 4: There is a conceptual weakness with the experiments using BC3 cells induced by chemical agents. According to the literature fewer than half of the cells will reactivate this virus, hence the majority of the cells will not activate MDA5/RIG1 in response to virus but in response to neighboring dying cells. This may explain why the biochemical purification failed to detect any virus derived RNAs.

The authors should demonstrate how many of the BC-3 cells and iSLK cells reactivate the virus.

The authors should demonstrate how this level of activation of the RIG-I MDA-5 pathway compares to natural activation in response to RNA virus infection or polyI:C treatment of the same cells.

(6)

Line 324: Doesn't RIG-I have higher affinity for RNA than MDA5 overall?

(7)

Figure 4 f, g: x Axis label is missing

(8)

Figure 4: Please comment on the p value differences between f and g. Is RIG-I still after adjustment for multiple comparison? Perhaps one would not expect RIG-I to be specific with regard to cellular pathways, since it binds smaller RNA fragments than MDA5.

(9)

Figure 5: An increase in vtRNAs by 2-fold is by substantially less than any change in viral RNAs after infection and unlikely to trigger a RIG-I response. Even though the result is statistically significant, the effect size may not have physiological relevance.

(10)

Line 409 probably means that vtRNAs were transfected AND CELLS WERE INDUCED? Otherwise it would mean that vtRNAs are activating this virus.

Figure 6A is confusing and may obfuscate the potentially very minor effect of treatment. Please use a more conventional scale using uninduced as mock set to 1 and the induced +/- RIG-I substrate as comparator. How were the significance statements arrived at? Are these technical or biological replicates?

(11)

Please quantify figure 6b and show single color panels as grayscale images.

(12)

Changing the order of panel c and b will allow the reader to compare directly the effect of vtRNA on viral and IFN gene expression.

(13)

The experiments in figure 6 should be repeated in uninduced cells.

Reviewer #2 (Remarks to the Author):

The manuscript entitled « RIG-I like receptor sensing of host RNAs facilitates the cell-intrinsic immune response to Kaposi's sarcoma-associated herpes virus infection » by Yang Zhao et al, shows that RIG-I and MDA5, two cytoplasmic sentinels, responsible for the detection of dsRNAs bearing non-self attributes, restrict KSHV lytic reactivation in various cell types. This is mediated by host cell mediated activation of these receptors. They also characterized the RNA PAMP ligands involved in RIG-I activation. Their model proposes that during KSHV lytic reactivation, the virus induced reduction of DUSP11 expression leads to the increase level of immunostimulatory

triphosphorylated vtRNA and activation of the RIG-I pathway and restriction of viral replication. In contrast in latent condition, DUSP11 prevents an accumulation of immunostimulatory triphosphorylated vtRNA and activation of RIG-I.

This is an incredible story described here. On the one hand there are increasing evidences of an endogenous source of cellular RNAs that have lost the attributes of self and are in fact detected by the cytoplasmic sentinels of the RLRs family. On the other hand, we have a DNA virus that spends a crazy energy with just unlikely strategies to target one of the members of this family, the RIG-I protein preventing its activation. This article makes it possible to reconcile the two sides of the same coin.

I am really excited about these results; the ms is very nice and interesting, the results are impressive and new. However, I have some specific comments and misunderstandings.

Figure 1.

Details.

Figure 1b, it would be nice to have the FACS analysis of figure 1b (at least the RFP) to be able to more directly compare the different ways to monitor viral restriction.

It would be nice to have the western blot analysis corresponding to the viral RNAs analyzed by qTR-PCR.

Please explicitly explain in the figure legend the significance of 24h hpi indicated in fig 1d (could be misleading)

More important

According to fig 1d, e, g, h, clearly MDA5 is much more active in the viral restriction than RIG-I, meaning that RIG-I appears to be quite inactive. If one considers that RIG-I, and a priori not MDA5, is the target of KSVH strategies to escape innate response during lytic infection, what does it mean? The RLRs involved in lytic infection and lytic reactivation are different? for a non-specialist, it could make sense knowing that the RIG-I inactivation is apparently mediated by tegument proteins, a priori absent in the initiation of reactivation?

The contradiction came from figure 1i. In this figure, IRF3 phosphorylation, as a hallmark of kinase activation downstream MAVS is at least as strongly inhibited in cells silenced for MDA5 than RIG-I, meaning that RIG-I is as active as MDA5 in viral induced activation of this pathway. Do the authors have an interpretation of this apparent contradiction? (MDA5 much more active in the viral restriction than RIG-I and RIG-I as active in blocking IRF3 phosphorylation than MDA5)

A possible track could be to monitor IFN stimulation at the level of type I or III mRNA level, meaning that one possible explanation would be that they both equally participate in IFN stimulation but IFN stimulation has nothing to do with the restriction. One way to discriminate between a "direct" versus an indirect (via secretion of IFN, activation of JAK-STAT pathway and activation of antiviral cellular functions) way of restriction of viral replication could be to silence IFN receptors or the JAK-STAT signaling pathway.

Figure 2

Details

Same remark, it would be nice to have the FACS analysis of figure 2a (at least the RFP).

Fig. 1b, as a non-expert, the restriction profiles appear to be different depending the viral gene considered (i.e. RIG-I and MDA5 appears to be equally active for ORF57 and 52 and MDA5 is much more active than RIG-I for PAN and vIL6). Does it mean that the restriction could affect different stages of the viral replication for MDA5 and RIG-I? is it just a kinetic issue?

I don't know how possible it is, but at one point, I would be interested in a direct comparison between an infection and a reactivation, in a silenced or an overexpression background.

Figure 4.

Fig. 4a. Intuitively I have the impression that there is a missing control in this experiment. If I understand correctly, mock means no transfection, IgG means immunoprecipitation from reactivated cells with control antibodies and J2 means immunoprecipitation from reactivated cells with antibodies against dsRNA and only the latter is active. I would be happy with a J2 control from cells latently infected cells (no reactivation) and from cells without any virus.

This control would also be interesting in fig.4c

Fig4c RIG-I very weakly binds RNA, authors explanation is that viral induced RIG-I deamidation has a negative effect on RNA binding. To my knowledge this virus induced deamidation is done initially by a tegument protein (UL37, a late protein) raising the questions: at what moment of the reactivation is this protein expressed? and is this protein active in the context of a reactivation? One possibility is to check deamidation on the immunoprecipitated RIG-I protein.

Figure 5

Concerning the in vitro made T7 vtRNAs, my feeling and my experience make me say that it is not specific of these RNAs. Basically, any T7 reaction will give the same result based on the fact that the T7 polymerase has the capacity to take the neo-synthesized RNA as a template to synthesize the complementary RNA (generating dsRNA with 5'ppp) (Replication of RNA by the DNA-dependent RNA polymerase of phage T7. Konarska MM, Sharp PA. Cell. 1989 May 5;57(3):423-31.PMID: 2720777). So, there is some work left if the goal is to make the point of the specificity.

What may appear strange is in fact that according to the data MDA5 is the main effector of the viral restriction and the authors have focused their study on the RIG-I RNA ligands.

Fig.5d,e Is this decrease level of mRNA coding for the phosphatase (DUSP11) specific of the reactivation? or is it also observed in a regular infection?

Line 403 to 405, "collectively, these experiments demonstrate that DUSP11 prevents an accumulation of immunostimulatory triphosphorylated vtRNA during KSHV lytic reactivation." I understand the sense of this sentence but for me it does not correspond to the message of the study, for me it is more "during KSHV lytic reactivation, the virus induced reduction of DUSP11 expression leads to the increase level of immunostimulatory triphosphorylated vtRNA and activation of the RIG-I pathway and restriction of viral replication...and in normal conditions, mock, latent..., DUSP11 effectively prevents an accumulation of immunostimulatory triphosphorylated vtRNA..."

One prediction that can be drawn from these results is that overexpression of DUSP11 should to some extent (maybe in the MDA5 silenced background) mimic the result obtained by silencing RIG-I. Maybe a good control.

Figure 6a and b same remark as for figure 5, for me there is no specificity there but it is not dramatic.

Reviewer: Dominique Garcin

Response to reviewer comments

We thank the editors and referees for their careful review of our manuscript and are encouraged by their excitement. We have revised the manuscript in response to the comments and provide a point-by-point response to each comment below. Within the manuscript our changes are highlighted in yellow.

Reviewer 1:

- Comment:** In addition to MDA5 and RIG-I, LGP2 also recognized foreign nucleic acids, but was not further assessed. Since Figure 1 has essentially been published before, it could be moved into supplemental data or enhanced with depletion of LGP2.

Response: To further enhance our data we have added additional FACS data, Fig. 1C & 1D, to quantify the effects of siRNA targeting MDA5, RIG-I, and MAVS on viral reactivation and infectious virion production. Considering there are no reported investigations into the role of MDA5 during KSHV infection, and we provide a direct comparison between RIG-I and MDA5, we believe this data is best presented in the main text.

- Comment:** The experiments are based on forced expression of MDA5 and RIG-I. This is necessary to purify interacting RNAs, but it introduces a bias in ascertaining the biological relevance with regard to viral replication and reactivation. Overexpression is not very relevant physiologically, as RIG-I and MDA5 need to become activated before initiating interferon signaling. Overexpression may artificially drive oligomerization.

Response: Prior to initiating this work we extensively tested several commercial RIG-I and MDA5 antibodies for their ability to immunoprecipitate either protein and their interacting RNAs, and no commercial antibodies were successful. Thus, as pointed out, expression of epitope-tagged MDA5 and RIG-I is necessary to identify interacting RNAs. The relevance of RIG-I and MDA5 is not asserted by only overexpression data, but also by siRNA depletion studies that indicate enhanced viral gene expression and replication upon MDA5, RIG-I, or MAVS depletion (Fig. 1, 3).

In regards to overexpression of RIG-I and MDA5 artificially driving interferon signaling, this cannot be tested in PEL cells because there is always a small percentage of cells undergoing spontaneous reactivation. Thus, to test this, we constructed additional doxycycline inducible FLAG-RIG-I and FLAG-MDA5 in uninfected HEK293 cells and tested whether overexpression drives interferon beta expression. We observed that overexpression of either FLAG-tagged protein is not sufficient to induce the expression of interferon beta by RT-qPCR. Importantly, we demonstrate that these constructed cell lines are responsive to RIG-I and MDA5 ligands, indicating they are capable of activating interferon beta. However we do not see an appropriate place within our text to describe this. Rather than include in Supplementary data we include it here for the reviewer to see.

3. **Comment:** Does the forced overexpressing of RIG-I or MDA5 result in greater levels of IRF3 phosphorylation or are the limiting factors of the IFN response MAVS and IRF3 levels?

Response: This data was originally described in Supplementary Figure 1, and now is present in Supplementary Figure 2f. There is a minor increase in IRF3 phosphorylation upon the expression of RIG-I and MDA5.

4. **Comment:** Figure 3: The WB analysis should include a late viral protein, which may show an even higher level of suppression by RIG-I or MDA5. In herpesviruses it is only after viral replication that all RNAs are transcribed including transcripts on opposite strands, which may give rise to double-stranded RNAs.

Response: ORF52, which is a well-established late gene, is quantified by RT-qPCR in Figure 3B and 3D. ORF52 was quantified by RT-qPCR as we were unable to obtain an antibody against it.

To further test this, we requested and received two additional antibodies against KSHV K8.1, a verified late viral protein, and vIRF1, suggested to be expressed with late kinetics in iSLK.219 cells in Arias C et al. 2014 (PLoS Pathogens). However, in multiple attempts the K8.1 antibody was extremely nonspecific and did not detect a single clear band. The vIRF1 antibody detected a single clear band and indeed this protein was significantly downregulated upon expression of RIG-I or MDA5. However, we detect slight vIRF1 expression in uninduced cells. Nonetheless, we have included the vIRF1 western blot data to Figure 3C.

5. **Comment:** Figure 4: There is a conceptual weakness with the experiments using BC3 cells induced by chemical agents. According to the literature fewer than half of the cells will reactivate this virus, hence the majority of the cells will not activate MDA5/RIG1 in response to virus but in response to neighboring dying cells. This may explain why the biochemical purification failed to detect any virus derived RNAs.

Response: We respectively disagree that the use of BC3 cells is a conceptual weakness and would suggest that this perhaps comes from a misinterpretation or lack of description in our text as well as how enrichment was determined. We have added to our results section to clarify that we observe robust lytic reactivation in our data and that lack of virus derived RNAs are specific to the fRIP-seq (Lines 347-348).

Additionally, in Supplementary Figure 3, we provide new bioinformatic analyses indicating that 48 h post-reactivation the number of sequencing reads detected per gene is on average higher for viral genes than for host genes. Thus, it would be more likely to enrich viral RNA than host, but that is not what we observe or report. Despite the increased reads for viral genes in the input libraries, only host transcripts are enriched. Furthermore, any RNA that is detected as expressed will have a fold enrichment score calculated. Thus, the percent of cells reactivating does not influence our ability to determine enrichment. Lastly, BC3 cells are a well-established system for investigating KSHV/PEL biology, and a setting with 100% PEL cell reactivation is not achievable by any condition nor would it be physiologically relevant.

6. **Comment:** The authors should demonstrate how many of the BC-3 cells and iSLK cells reactivate the virus.

Response: We now include FACS to show iSLK reactivation in Fig. 1, and PAN RNA FISH-FLOW for BC3 reactivation is presented in Supplementary Figure S2.

7. **Comment:** The authors should demonstrate how this level of activation of the RIG-I MDA-5 pathway compares to natural activation in response to RNA virus infection or polyI:C treatment of the same cells.

Response: As suggested, we have microporated poly I:C into PEL cells and analyzed IRF3 phosphorylation 12 h post transfection and compared this to KSHV lytic reactivation at the same time point. This data is now presented in Supplementary Figure S2.

8. **Comment:** Line 324: Doesn't RIG-I have higher affinity for RNA than MDA5 overall?

Response: Considering RIG-I and MDA5 have different requirements for binding RNA this is something that has not been addressed in the field and is likely not possible.

9. **Comment:** Figure 4 f, g: x Axis label is missing

Response: Thank you for pointing out this omission. We have now labeled the X axis.

10. **Comment:** Figure 4: Please comment on the p value differences between f and g. Is RIG-I still after adjustment for multiple comparison? Perhaps one would not expect RIG-I to be specific with regard to cellular pathways, since it binds smaller RNA fragments than MDA5.

Response: Thank you for pointing out this omission. The p-values shown are adjusted p-values and take into account multiple comparison. We have corrected the figure to indicate that the p-values described are adjusted.

11. **Comment:** Figure 5: An increase in vtRNAs by 2-fold is by substantially less than any change in viral RNAs after infection and unlikely to trigger a RIG-I response. Even though the result is statistically significant, the effect size may not have physiological relevance.

Response: It is not the change in vtRNA abundance as suggested, but rather the accumulation of triphosphorylated vtRNAs, as well as other identified triphosphorylated RNAs (e.g. Y RNAs, and Alu RNAs; lines 482-488, and 495-497) that drive the RIG-I response. Additionally, we demonstrate vtRNAs isolated from lytic PEL cells are indeed immunostimulatory in a CIP-dependent manner in Figure 5H.

12. **Comment:** Line 409 probably means that vtRNAs were transfected AND CELLS WERE INDUCED? Otherwise it would mean that vtRNAs are activating this virus.

Response: Yes, thank you for bringing this to our attention. We have now reworded the text accordingly.

13. **Comment:** Figure 6A is confusing and may obfuscate the potentially very minor effect of treatment. Please use a more conventional scale using uninduced as mock set to 1 and the induced +/- RIG-I substrate as comparator. How were the significance statements arrived at? Are these technical or biological replicates?

Response: Thank you for pointing this out. The Y-axis was incorrectly labeled in our initial submission, leading to the appearance of a minor effect. We have corrected the Y-axis and it now reads relative RNA levels. Now with the correct labeling you can observe that the effect is not minor and is in fact quite large. All experiments were performed in triplicates and represent biological replicates.

14. Comment: Please quantify figure 6b and show single color panels as grayscale images.

Response: We now provide quantification of the RFP positive cells; this figure is now Fig. 6A with the reorganization as suggested in the below comment. Given that the whole manuscript has shown GFP and RFP in green and red, respectively, we have left the coloring of the images.

15. Comment: Changing the order of panel c and b will allow the reader to compare directly the effect of vtRNA on viral and IFN gene expression.

Response: Thank you for this suggestion. We have reorganized Figure 6. The GFP and RFP fluorescent images and quantification are now presented as Fig. 6a and b, and the RT-qPCR gene expression measurements are presented as Fig. 6c and 6d.

16. The experiments in figure 6 should be repeated in uninduced cells.

Response: We have now included this data in Figure 6, and Supplementary Figure 9.

Reviewer 2:

1. Comment: Figure 1b, it would be nice to have the FACS analysis of figure 1b (at least the RFP) to be able to more directly compare the different ways to monitor viral restriction.

Response: We have now added FACS for the RFP as suggested.

2. Comment: It would be nice to have the western blot analysis corresponding to the viral RNAs analyzed by qTR-PCR.

Response: We were only successful in obtaining a few antibodies against KSHV proteins—these are the proteins which we probed by western blot in Figure 1. However, we have now added additional RT-qPCR quantifications of the proteins that we detected by western blot. This is now included in Supplementary Figure S1.

3. Comment: Please explicitly explain in the figure legend the significance of 24h hpi indicated in fig 1d (could be misleading)

Response: Thank you for pointing this out. We have not removed the “hpi” labeling from the manuscript.

4. Comment: According to fig 1d, e, g, h, clearly MDA5 is much more active in the viral restriction than RIG-I, meaning that RIG-I appears to be quite inactive. If one considers that RIG-I, and a priori

not MDA5, is the target of KSVH strategies to escape innate response during lytic infection, what does it mean? The RLRs involved in lytic infection and lytic reactivation are different? for a non-specialist, it could make sense knowing that the RIG-I inactivation is apparently mediated by tegument proteins, a priori absent in the initiation of reactivation?

Response: Indeed, we do not observe a prominent role for the RLRs during de novo infection (Supplementary Figure 1), suggesting the innate responses during de novo infection are distinct from reactivation.

We do observe that RIG-I is partially inactivated during lytic infection. As a result of your below comment (Comment 9), we investigated RIG-I deamidation in lytic PEL cells and observed deamidation. This is presented in Supplementary Figure 2.

Additionally, as suggested, we have expanded our comments regarding RIG-I inactivation during de novo infection by tegument proteins in the introduction (lines 65-67).

5. **Comment:** Figure 1i. In this figure, IRF3 phosphorylation, as a hallmark of kinase activation downstream MAVS is at least as strongly inhibited in cells silenced for MDA5 than RIG-I, meaning that RIG-I is as active as MDA5 in viral induced activation of this pathway. Do the authors have an interpretation of this apparent contradiction? (MDA5 much more active in the viral restriction than RIG-I and RIG-I as active in blocking IRF3 phosphorylation than MDA5)

Response: This a keen observation, and in fact one that we are pursuing in a separate project that is out of the scope of this current work. However, we now included a new paragraph within the discussion regarding this (lines 454 – 463). Briefly, our observation, coupled with recent literature describing RLR-dependent, but MAVS-independent, STAT activation, as well as STING-mediated restriction of RNA viruses that is independent of IFN stimulation, paint novel and broad roles for nucleic acid receptors in host defense.

6. **Comment:** Same remark, it would be nice to have the FACS analysis of figure 2a (at least the RFP). Fig. 1b, as a non-expert, the restriction profiles appear to be different depending the viral gene considered (i.e. RIG-I and MDA5 appears to be equally active for ORF57 and 52 and MDA5 is much more active than RIG-I for PAN and vIL6). Does it mean that the restriction could affect different stages of the viral replication for MDA5 and RIG-I? is it just a kinetic issue?

Response: We have now added FACS analysis of RFP to Figure 2.

The effects on viral gene expression does not fit within a purely kinetic framework. We have observed differential transcriptional regulation of viral genes using nascent RNA profiling. However, that work is both out of the scope of this study and premature.

7. **Comment:** I don't know how possible it is, but at one point, I would be interested in a direct comparison between an infection and a reactivation, in a silenced or an overexpression background.

Response: We now provide additional data testing whether siRNA-depletion of RIG-I, MAVS, or MDA5 impacts de novo infection into uninfected iSLK cells. We observe a minor increase in viral infectivity upon depletion of the RNA sensors or adapter MAVS. This data is now included as Supplementary Figure 1. This indicates that while the RLRs play a prominent role in during viral reactivation, their role during de novo infection is less significant.

8. **Comment:** Fig. 4a. Intuitively I have the impression that there is a missing control in this experiment. If I understand correctly, mock means no transfection, IgG means immunoprecipitation from reactivated cells with control antibodies and J2 means immunoprecipitation from reactivated cells with antibodies against dsRNA and only the latter is active. I would be happy with a J2 control from cells latently infected cells (no reactivation) and from cells without any virus. This control would also be interesting in fig.4c

Response: We have added the requested controls for the J2 immunoprecipitation for Fig. 4a, and lines 337-339.

Regarding Figure 4C, while our manuscript was under review we discovered that MDA5 itself can be radioactively labeled by PNK and ATP in vitro, confounding the direct comparison between RIG-I and MDA5 using radioactive labeling. Because this we have removed this panel.

9. **Comment:** Fig4c RIG-I very weakly binds RNA, authors explanation is that viral induced RIG-I deamidation has a negative effect on RNA binding. To my knowledge this virus induced deamidation is done initially by a tegument protein (UL37, a late protein) raising the questions: at what moment of the reactivation is this protein expressed? and is this protein active in the context of a reactivation? One possibility is to check deamidation on the immunoprecipitated RIG-I protein.

Response: Although we have removed Fig4C, we have now quantified the expression of ORF75, the KSHV protein which has been demonstrated to facilitate deamidation of RIG-I. ORF75 expression is highest on 48 h post lytic induction—which is when our fRIP-seq experiments were performed. This is now shown in Supplementary Figure 2. Additionally, we have verified RIG-I deamidation in lytic PEL cells by mass spectrometry. This data is also shown in Supplementary Figure 2.

10. **Comment:** Figure 5, Concerning the in vitro made T7 vtRNAs, my feeling and my experience make me say that it is not specific of these RNAs. Basically, any T7 reaction will give the same result based on the fact that the T7 polymerase has the capacity to take the neo-synthesized RNA as a template to synthesize the complementary RNA (generating dsRNA with 5'ppp)(Replication of RNA by the DNA-dependent RNA polymerase of phage T7. Konarska MM, Sharp PA. Cell. 1989 May 5;57(3):423-31.PMID: 2720777). So, there is some work left if the goal is to make the point of the specificity.

Response: Thank you for pointing out this omission within the methods. First, the in vitro transcription reactions were fractionated and only full-length vtRNAs were gel excised and purified. This is now included in the methods (Lines 176-178). This rules out the concern of dsRNA formation from aberrant short antisense RNAs or larger unintended products produced in the transcription reaction (as in the referenced manuscript). In fact, as observed in the transcription reaction in Figure 5b, there are no other abundant RNAs produced.

To rule out that an aberrant antisense RNA of exact or similar size was synthesized and annealed to the vtRNAs producing the immunostimulatory dsRNA, vtRNAs were treated with RNase T1, a single stranded RNase specific for G residues. As shown in Supplementary Figure 5, RNase T1 treatment degrades the vtRNAs, indicating the RNA is not bound to an antisense transcribed RNA.

11. **Comment:** What may appear strange is in fact that according to the data MDA5 is the main effector of the viral restriction and the authors have focused their study on the RIG-I RNA ligands.

Response: We have added to the manuscript a sentence to offer an explanation as to why we focus on RIG-I ligands in this manuscript (line 370). We are currently pursuing a separate MDA5-related story.

12. Comment: Fig.5d,e Is this decrease level of mRNA coding for the phosphatase (DUSP11) specific of the reactivation? or is it also observed in a regular infection?

Response: We have performed de novo infections into iSLK cells (which lack KSHV) and quantified expression of DUSP11 by RT-qPCR at 6, 12, 24, and 48 h post-infection. We do not observe any significant change in DUSP11 expression following a de novo infection and the establishment of latency. This data is now presented in Supplementary Figure 6.

13. Comment: Line 403 to 405, “collectively, these experiments demonstrate that DUSP11 prevents an accumulation of immunostimulatory triphosphorylated vtRNA during KSHV lytic reactivation.” I understand the sense of this sentence but for me it does not correspond to the message of the study, for me it is more “during KSHV lytic reactivation, the virus induced reduction of DUSP11 expression leads to the increase level of immunostimulatory triphosphorylated vtRNA and activation of the RIG-I pathway and restriction of viral replication...and in normal conditions, mock, latent..., DUSP11 effectively prevents an accumulation of immunostimulatory triphosphorylated vtRNA...”

Response: Thank you for this suggestion. We have now reworded our text to more clearly state the message of our study.

14. Comment: Figure 6a and b same remark as for figure 5, for me there is no specificity there but it is not dramatic.

Response: We have addressed the issue of in vitro transcribed RNA in response #10.

Reviewers' comments:

Reviewer #2 (Remarks to the Author):

Revision, RIG-I like receptor sensing of host RNAs facilitates the cell-intrinsic immune response to Kaposi's sarcoma-associated herpesvirus infection, Yang Zhao, Xiang Ye, Will Dunker, Yu Song John Karijolich.

Having carefully read the revised version of the article, I'm fully convinced and satisfied by the responses made to my comments.

Concerning the response made to the first comment of reviewer 1, I'm not sure I understand the response. The comment is mainly related to LGP2 as a regulator of RIG-I and MDA5 and its possible role in this story. It is a remarkably interesting comment. LGP2 has a positive effect on MDA5 activation and a negative effect on RIG-I (and possibly on RIG-I activation by self-RNAs). Consequently, the result of silencing LGP2 is potentially very interesting, pointing for the physiological relative importance of MDA5 or RIG-I in KSHV reactivation. The answer completely obscures this part of the comment. What I can understand is, it could be outside the scope of this study but it remains a very interesting comment.

I have a last comment. The first part of the paper is related to the restriction exerted by RIG-I and MDA5 on KSHV reactivation. Knowing that RIG-I and MDA5 activation is done upon binding with dsRNAs, the authors show that they mainly bind host derived RNAs during KSHV lytic reactivation. They show that cellular vault RNAs is associated with RIG-I. The immunostimulatory part of vtRNAs is dependent on the presence of free 5'ppp due to a decrease of DUSP11 expression during KSHV reactivation. They finally show that 5'ppp-vtRNAs are sufficient to prevent KSHV lytic reactivation, closing the loop. I repeat this is a remarkable work and the fact that a cellular activity (DUSP11) affecting host derived RNA processing have an impact on innate immune response is highly interesting. This is another example of a cellular pathway for the production of endogenous RNA PAMPs triggering RLR activation.

In my mind, the activation of RIG-I or MDA5 leads to the activation of type 1 or 3 IFN expression as well as other cytokines. These cytokines are secreted and exert their effects in Cis- or Trans- by binding to their respective receptors. IFN leading to the activation of ISGs. No experiment is made here to demonstrate that the restriction is exerted via IFN expression (IFN receptor silencing, STAT1 silencing ...) or in a direct way. Nevertheless, in the model presented in figure 6 e (I did not find in the text where this figure was mentioned), one has the impression that the effect of the restriction is direct via the phosphorylation of IRF3 which would have a direct effect on the expression of ISGs. I think this can be misleading and deserves to be nuanced.

We thank the editor and the referee for their enthusiasm regarding our manuscript. Below we address each comment.

1. **Comment:** Concerning the response made to the first comment of reviewer 1, I'm not sure I understand the response. The comment is mainly related to LGP2 as a regulator of RIG-I and MDA5 and its possible role in this story. It is a remarkably interesting comment. LGP2 has a positive effect on MDA5 activation and a negative effect on RIG-I (and possibly on RIG-I activation by self-RNAs). Consequently, the result of silencing LGP2 is potentially very interesting, pointing for the physiological relative importance of MDA5 or RIG-I in KSHV reactivation. The answer completely obscures this part of the comment. What I can understand is, it could be outside the scope of this study but it remains a very interesting comment.

Response: We apologize for not fully addressing comment 1 from reviewer 1. Indeed, we agree that an investigation into the role of LGP2 in KSHV infection is potentially very interesting, especially since LGP2 can bind RNA. However, determining whether LGP2 is involved in KSHV infection, as well as the potential mechanism, is out of the scope of our current manuscript and can be pursued in future follow up studies.

2. **Comment:** The model presented in figure 6e (I did not find in the text where this figure was mentioned), one has the impression that the effect of the restriction is direct via the phosphorylation of IRF3 which would have a direct effect on the expression of ISGs. I think this can be misleading and deserves to be nuanced.

Response: Thank you for pointing out the omission of Fig. 6e in the text. We now mention Fig. 6e on line 453.

Thank you for the comment that Fig. 6e could be potentially misleading as it suggests IRF3 phosphorylation has a direct effect on viral restriction. To clarify our figure we have now modified the lytic portion of Fig. 6e, as well as modified the text of our figure legend. Specifically, we have removed the viral restriction schematic. Our figure now depicts that a reduction in DUSP11 results in an accumulation of triphosphorylated RNAs that lead to an IRF3 phosphorylation and an antiviral gene expression program--rather than a direct antiviral effect.

REVIEWERS' COMMENTS:

Reviewer #2 (Remarks to the Author):

I am satisfied with the answers provided as well as with the modification of Figure 6E and I continue to think that this is an important study on the ability of the cell to produce its own RNA PAMPs.

RESPONSE TO REVIEWERS' COMMENTS:

Reviewer #2 (Remarks to the Author):

I am satisfied with the answers provided as well as with the modification of Figure 6E and I continue to think that this is an important study on the ability of the cell to produce its own RNA PAMPs.

Comment: Thanks.